# Distinct states of nucleolar stress induced by anticancer drugs

Tamara A Potapova[1]*, Jay R Unruh[1], Juliana Conkright-Fincham[1], Charles AS Banks[1], Laurence Florens[1], David Alan Schneider[2], Jennifer L Gerton[1,3]

[1]Stowers Institute for Medical Research, Kansas City, United States; [2]Department of Biochemistry and Molecular Genetics, University of Alabama at Birmingham, Birmingham, United States; [3]Department of Biochemistry and Molecular Biology, University of Kansas Medical Center, Kansas City, United States

*For correspondence:
tpo@stowers.org

Competing interest: The authors declare that no competing interests exist.

**Abstract** Ribosome biogenesis is a vital and highly energy-consuming cellular function occurring primarily in the nucleolus. Cancer cells have an elevated demand for ribosomes to sustain continuous proliferation. This study evaluated the impact of existing anticancer drugs on the nucleolus by screening a library of anticancer compounds for drugs that induce nucleolar stress. For a readout, a novel parameter termed 'nucleolar normality score' was developed that measures the ratio of the fibrillar center and granular component proteins in the nucleolus and nucleoplasm. Multiple classes of drugs were found to induce nucleolar stress, including DNA intercalators, inhibitors of mTOR/PI3K, heat shock proteins, proteasome, and cyclin-dependent kinases (CDKs). Each class of drugs induced morphologically and molecularly distinct states of nucleolar stress accompanied by changes in nucleolar biophysical properties. In-depth characterization focused on the nucleolar stress induced by inhibition of transcriptional CDKs, particularly CDK9, the main CDK that regulates RNA Pol II. Multiple CDK substrates were identified in the nucleolus, including RNA Pol I– recruiting protein Treacle, which was phosphorylated by CDK9 in vitro. These results revealed a concerted regulation of RNA Pol I and Pol II by transcriptional CDKs. Our findings exposed many classes of chemotherapy compounds that are capable of inducing nucleolar stress, and we recommend considering this in anticancer drug development.

## eLife assessment

This study and associated data is **compelling**, novel, **important**, and well-carried out. The study demonstrates a novel finding that different chemotherapeutic agents can induce nucleolar stress, which manifests with varying cellular and molecular characteristics. The study also proposes a mechanism for how a novel type of nucleolar stress driven by CDK inhibitors may be regulated. The study sheds light on the importance of nucleolar stress in defining the on-target and off-target effects of chemotherapy in normal and cancer cells.

## Introduction

The nucleolus is the most prominent nuclear organelle. Its primary function is the biogenesis of ribosomes – a pivotal housekeeping process essential for the translation of all proteins. Ribosome biogenesis is a major metabolic expense in a cell. This biosynthetic program requires transcription and processing of the most abundant cellular RNA – the ribosomal RNA (rRNA), and the production of 80 ribosomal proteins and hundreds of other nucleolar proteins involved in rRNA processing and assembly of ribosomal subunits (*Moss and Stefanovsky, 2002*; *Granneman and Tollervey, 2007*). Rapidly proliferating cancer cells have ribosome biogenesis shifted into overdrive, which may be one

**eLife digest** Ribosomes are cell structures within a compartment called the nucleolus that are required to make proteins, which are essential for cell function. Due to their uncontrolled growth and division, cancer cells require many proteins and therefore have a particularly high demand for ribosomes. Due to this, some anti-cancer drugs deliberately target the activities of the nucleolus. However, it was not clear if anti-cancer drugs with other targets also disrupt the nucleolus, which may result in side effects.

Previously, it had been difficult to study how nucleoli work, partly because in human cells they vary naturally in shape, size, and number. Potapova et al. used fluorescent microscopy to develop a new way of assessing nucleoli based on the location and ratio of certain proteins. These measurements were used to calculate a "nucleolar normality score".

Potapova et al. then tested over a thousand anti-cancer drugs in healthy and cancerous human cells. Around 10% of the tested drugs changed the nucleolar normality score when compared to placebo treatment, indicating that they caused nucleolar stress. For most of these drugs, the nucleolus was not the intended target, suggesting that disrupting it was an unintended side effect.

Drugs inhibiting proteins called cyclin-dependent kinases caused the most drastic changes in the size and shape of nucleoli, disrupting them completely. These kinases are known to be involved in activating enzymes required for general transcription. Potapova et al. showed that they also are involved in production of ribosomal RNA, revealing an additional role in coordinating ribosome assembly.

Taken together, the findings suggest that evaluating the effect of new anti-cancer drugs on the nucleolus could help to develop future treatments with less toxic side effects. The experiments also reveal new avenues for researching how cyclin-dependent kinases control the production of RNA more generally.

of their primary metabolic alterations (*Drygin et al., 2010*). Several anticancer drugs targeting ribosome biogenesis pathways have been developed (*Ferreira et al., 2020*), yet anticancer therapies targeting nucleolar function have not been a major focus of new drug development because of the universal role of this pathway in maintaining basic cellular functions. The main objective of this study was to identify the compounds that disrupt normal nucleolar physiology and further explore the new and unconventional agents that induce nucleolar stress.

The nucleolus is a membrane-less organelle that assembles around ribosomal RNA genes (rDNA). rRNA genes in eukaryotic cells are present in hundreds of tandemly arranged repetitive copies that are transcribed by RNA polymerase I (Pol I) (reviewed in *Potapova and Gerton, 2019*). Nucleolar anatomy in animal cells is comprised of three distinct compartments: the fibrillar center (FC), the dense fibrillar component (DFC), and the granular component (GC) (*Pederson, 2011*). FC is the site of transcription that consists of rDNA and its associated transcription machinery such as transcription factor UBF and RNA Pol I. The DFC is the site of pre-rRNA processing distinguished by early RNA processing factors such as fibrillarin. The GC contains proteins involved in late rRNA processing and assembly of pre-ribosomal particles. It is marked by proteins such as nucleolin and nucleophosmin (NPM1). Changes in nucleolar organization during stress have not been studied extensively, except for the inhibition of RNA Pol I that causes the reorganization of rDNA arrays and associated FC proteins into round nucleoli with peripheral 'stress caps.' Biophysical and biochemical events underlying nucleolar reorganization under stress remain poorly understood.

Nucleoli in mammalian cells can be highly polymorphic – different in shape, size, and number. It is difficult to find a single parameter that can quantitatively distinguish normal nucleolar anatomy from abnormal. To quantify the impact of anticancer drugs on nucleoli, we developed a novel imaging-based parameter that we termed 'the nucleolar normality score.' It is based on measuring nucleolar/nucleoplasmic ratios of GC component nucleolin and FC component UBF. Measuring the normality score allowed us to detect distinct states of nucleolar stress in a screen of more than a thousand chemical compounds developed as anticancer agents. The screen was conducted using a noncancer-derived cell line RPE1. This cell line was selected for evaluating the effects of anticancer drugs on normal nucleolar function. The outcome of the screen provided a broad atlas of aberrant nucleolar

morphologies and their molecular triggers, where multiple drugs with the same target often produced a similar morphological and functional state.

We classify four distinct categories of nucleolar stress: (1) canonical nucleolar stress with the formation of stress caps caused by DNA intercalators, (2) metabolic suppression of function caused by PI3K and mTOR inhibitors, (3) proteotoxicity with or without formation of aggresomes caused by HSP90 and proteasome inhibitors, and (4) nucleolar dissolution with an extended bare rDNA scaffold caused by cyclin-dependent kinase (CDK) inhibitors. An in-depth examination of the nucleolar stress caused by CDK inhibitors uncovered previously unknown regulation of RNA Pol I by CDKs and suggests the possibility of concerted regulation of Pol I and Pol II by transcriptional CDK activity. Finally, our study highlights the fact that many anticancer drugs can cause unintended effects on the nucleolus that can underlie off-target toxicity, which should be considered in the development and use of antineoplastic agents.

## Results

### The biological basis for the nucleolar normality score

To establish a robust quantitative method for measuring nucleolar stress, we first investigated the properties of nucleolar components during the inhibition of RNA Pol I. Inhibition of Pol I transcription manifests in acute morphological changes referred to as canonical nucleolar stress. Canonical nucleolar stress is well characterized in the instance of antineoplastic agent actinomycin D (dactinomycin) that stalls Pol I transcription by intercalating into G/C-rich rDNA. This causes nucleoli to shrink and round up, with the partial dissolution of some GC proteins into the nucleoplasm and the formation of so-called 'stress caps' at the nucleolar periphery. Stress caps consist of segregated rDNA with bound FC proteins (*Shav-Tal et al., 2005*; *Mangan et al., 2017*). In this study, we inhibited Pol I using a small molecule compound CX-5461 (*Drygin et al., 2011*). This drug has been shown to arrest Pol I at the rDNA promoter, which blocks transcription initiation (*Mars et al., 2020*).

To quantify the effects of CX-5461 on nucleoli by live imaging, we used hTERT immortalized human RPE1 cell lines stably expressing GC component nucleolin tagged with GFP, or FC component UBF tagged with the GFP. Expression of eGFP-nucleolin enabled us to visualize the process of nucleolar shrinking and rounding up, and the formation of small circular remnants within the first hour after RNA Pol I inhibitor treatment (*Figure 1A* and *Video 1*). With the first hour after treatment, the average intensity of eGFP-nucleolin decreased in the nucleoli and increased in the nucleoplasm (*Figure 1A*, right panel), indicating a higher proportion of total nucleolin dissolved in the nucleoplasm. This resulted in a decrease in the fluorescence intensity ratio of the nucleolar pool relative to the nucleoplasmic pool. In cells expressing eGFP-UBF, treatment with CX-5461 induced UBF condensation at the periphery of the nucleolar remnants and the formation of stress caps (*Figure 1B* and *Video 2*). The intensity of eGFP-UBF increased in these small stress caps, while the intensity in the nucleoplasm did not change (*Figure 1B*, right panel). For eGFP-UBF, the average fluorescence intensity ratio of the stress caps relative to the nucleoplasmic pool increased.

Next, we investigated the mobility of the eGFP-nucleolin and eGFP-UBF by fluorescence recovery after photobleaching (FRAP) before and after nucleolar stress induced with CX-5461. Nucleolin became more mobile in stressed cells (the average half-time recovery $T_{1/2}$ went down from 4.58 ± 1.88 s to 2.89 ± 0.88 s, *Figure 1C*), consistent with its redistribution to the nucleoplasm. The $T_{1/2}$ of UBF did not significantly change with stress and stayed on the average of 12–14 s (*Figure 1D*), indicating that the rDNA-binding properties of UBF that likely underlie its FRAP behavior were not affected by RNA Pol I inhibition. This is consistent with UBF acting as a stable bookmark of the rDNA during mitosis, when RNA Pol I activity is very low (*Roussel et al., 1993*; *Gébrane-Younès et al., 1997*).

This contrasting behavior of nucleolin and UBF after Pol I inhibition provided the basis for the nucleolar stress parameter that we termed the nucleolar normality score. The nucleolar normality score is a ratio of the nucleolar fraction of nucleolin relative to the nucleolar fraction of UBF (*Figure 1E*). Image processing and calculation of the normality score are explained in detail in 'Materials and methods.' This parameter is applicable to fixed cells where both proteins are labeled by immunofluorescence. In a normal, unstressed situation the average normality score has a consistent value that is characteristic for a given experimental system. As nucleolin dissolves in the nucleoplasm and UBF becomes segregated, the normality score decreases. The normality score was very robust at detecting the strong

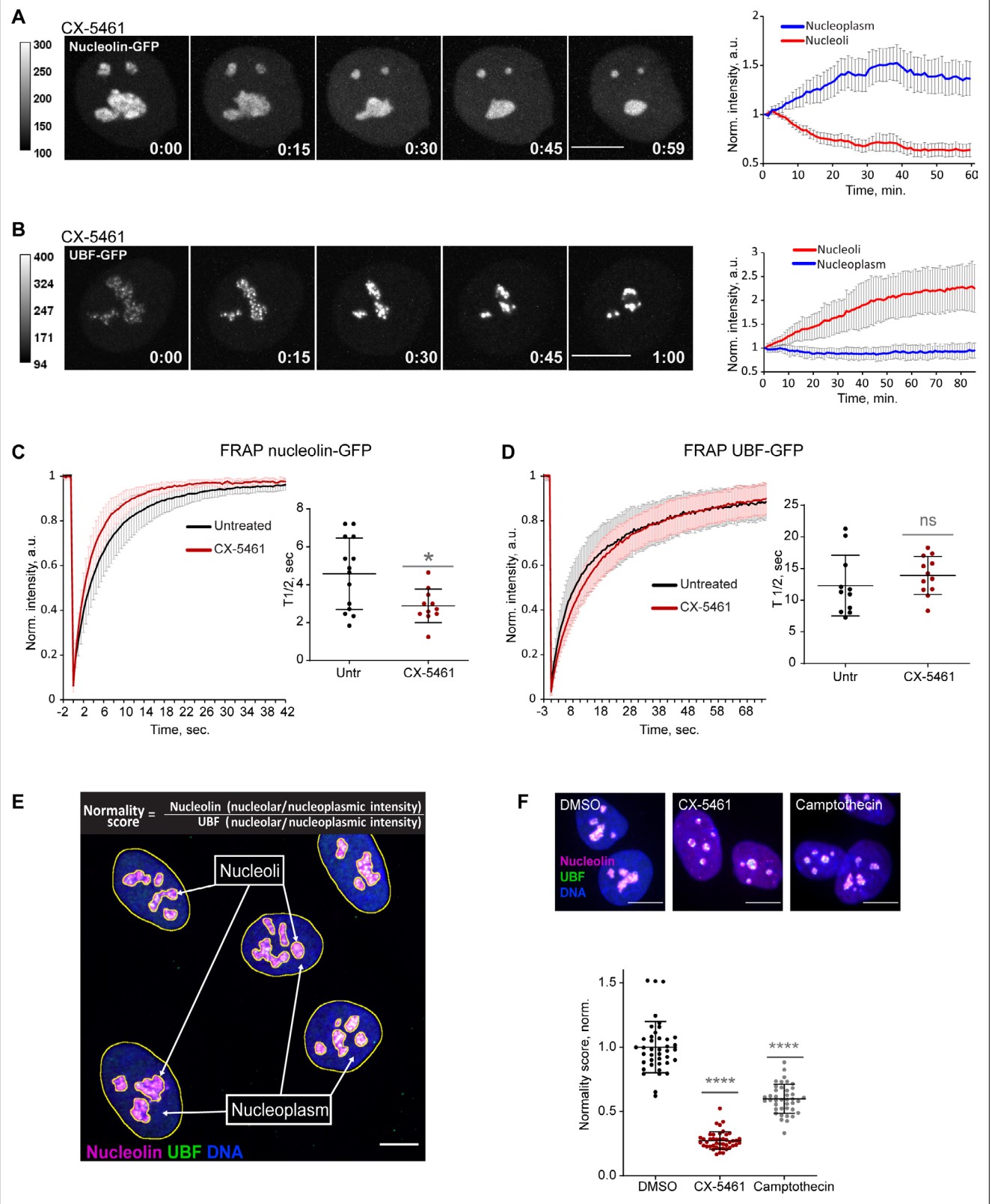

**Figure 1.** Nucleolar normality score as a parameter for measuring nucleolar stress. (**A**) Time-lapse images of eGFP-nucleolin expressing cell treated with 2.5 µM Pol I inhibitor CX-5461 at time 0 are shown. Nucleoli shrink and round up forming small circular remnants. Fluorescence intensity, indicated by the heatscale, decreases in nucleolar remnants and increases in the nucleoplasm. The complete video sequence is shown in *Video 1*. Bar, 10 µm. The plot on the right shows the average intensity of eGFP-nucleolin in nucleoli and in the nucleoplasm normalized to the initial intensity at time 0. The

*Figure 1 continued on next page*

*Figure 1 continued*

plot is an average of 10 cells, bars denote standard deviation. (**B**) Time-lapse images of eGFP-UBF expressing cell treated with 2.5 μM CX-5461 at time 0 are shown. UBF condenses on the periphery of nucleolar remnants forming stress caps of high fluorescence intensity. The complete video sequence is shown in *Video 2*. Bar, 10 μm. The plot on the right shows the average intensity of eGFP-UBF in stress caps and in the nucleoplasm normalized to the initial intensity at time 0. The plot is an average of 13 cells, bars denote standard deviation. (**C**) Fluorescence recovery after photobleaching (FRAP) analysis of eGFP-nucleolin in untreated cells and cells treated with 2.5 μM CX-5461 is shown. The plot is an average of normalized fluorescence intensities of 14 and 10 cells. Bars denote standard deviation. The graph on the right shows corresponding individual $T_{1/2}$ measurements. Asterisk indicates p<0.05 (*t*-test comparing the drug-treated group to untreated). (**D**) FRAP analysis of eGFP-UBF in untreated cells and cells treated with 2.5 μM CX-5461. The plot is an average of normalized fluorescence intensities of 11 and 12 cells. Bars denote standard deviation. The graph on the right shows corresponding individual $T_{1/2}$ measurements. *t*-test did not detect a significant difference between the two treatments. (**E**) The immunofluorescence image illustrates the Nucleolar Normality score measurement. RPE1 cells were labeled with antibodies against nucleolin and UBF and counterstained with DAPI. Segmentation of nucleolar regions was performed on UBF, and whole nuclei were segmented on DAPI. Nucleoplasm regions are areas within the nuclei without nucleoli. The nucleoplasmic intensity was calculated by subtracting the integrated intensity of nucleoli from the integrated intensity of the whole nuclei. For both nucleolin and UBF, the integrated intensity of the nucleolar regions of each cell was divided by the integrated intensity of the nucleoplasm of that cell, giving the nucleolar/nucleoplasm ratio. Dividing the nucleolar/nucleoplasm ratio of the nucleolin by the nucleolar/nucleoplasm ratio of the UBF provides a nucleolar normality score for each cell. (**F**) Normality score measurements of individual cells treated with DMSO (vehicle), 2.5 μM CX-5461, or 5 μM topoisomerase inhibitor camptothecin, normalized to the average value of DMSO-treated cells. More than 40 individual cells were measured for each condition. Asterisks indicate p<0.0001 (unpaired *t*-test comparing drug-treated groups to DMSO).

The online version of this article includes the following source data for figure 1:

**Source data 1.** Source data for *Figure 1A-F*.

nucleolar stress phenotype caused by CX-5461, but it was also proven to detect more subtle morphological changes, such as the stress caused by topoisomerase inhibitor camptothecin (*Figure 1F*). This parameter allowed us to detect nucleolar stress phenotypes that are less pronounced and measure the degree of nucleolar perturbations of various origins.

## High-throughput imaging screen for anticancer drugs that induce nucleolar stress

We screened nucleolar normality in cells treated with a chemical library containing 1180 anticancer compounds developed for multiple cancers, some of them FDA-approved and used clinically. The main goal was to broadly identify and categorize distinct states of nucleolar stress and their molecular triggers. For the screen, normal human hTERT-immortalized RPE1 cells were seeded in 384-well plates and treated with the library compounds at 1 μM and 10 μM for 24 hr. Drug treatment was followed by fixation and labeling with antibodies against UBF and nucleolin (*Figure 2A*). Forty single-plane fields containing hundreds of cells were imaged per well. Compounds were called hits if their normality score was more than 2 standard deviations away from the DMSO (vehicle) control average. Of 1180 compounds present in the library, 12.9% were hits. Also, 7% of the compounds in the library were hits at both 1 and 10 μM, and 5.8% were hits only at 10 μM (*Figure 2B*). The majority of the hits were

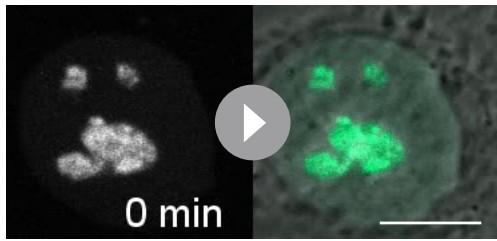

**Video 1.** Fluorescence and phase-contrast time-lapse video of a human RPE1 cell stably expressing eGFP-nucleolin that was treated with 2.5 μM RNA Pol I inhibitor CX-5461. Nucleoli shrink and round up forming small circular remnants. Fluorescence intensity decreases in nucleolar remnants and increases in the nucleoplasm. Time is indicated as minutes after drug addition. Bar, 10 μm.

https://elifesciences.org/articles/88799/figures#video1

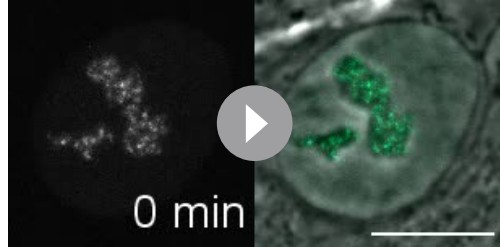

**Video 2.** Fluorescence and phase-contrast time-lapse video of a human RPE1 cell stably expressing eGFP-UBF that was treated with 2.5 μM RNA Pol I inhibitor CX-5461. UBF condenses on the periphery of nucleolar remnants forming stress caps of high fluorescence intensity. Time is indicated as minutes after drug addition. Bar, 10 μm.

https://elifesciences.org/articles/88799/figures#video2

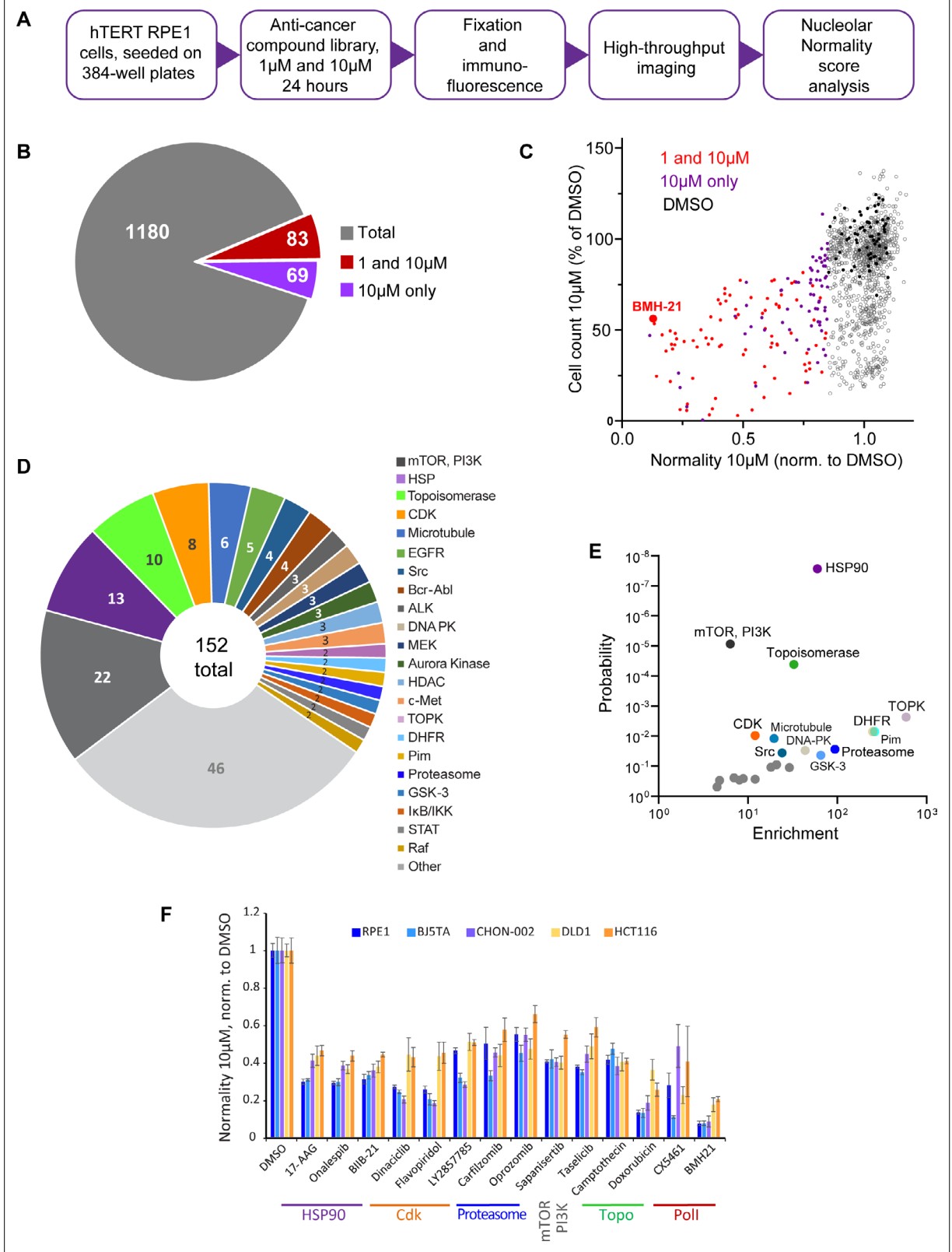

**Figure 2.** Anticancer drug screen for compounds that induce nucleolar stress. (**A**) The diagram illustrates the workflow for the library screen for anticancer compounds that induce nucleolar stress. (**B**) From the total 1180 compounds, 83 were hits at both 1 and 10 µM, and 69 were hits at 10 µM only. The full list of hit compounds with normality scores is provided in **Supplementary file 1**. (**C**) Normality score results from cells treated with 10 µM drug are plotted versus cell count. Both parameters were normalized to the average of the DMSO control (black points). Red points denote hits at 1

*Figure 2 continued on next page*

*Figure 2 continued*

and 10 µM, purple points 10 µM only. BMH21 is a Pol I inhibitor present in the library and serves as an internal control. (**D**) Combined 1 and 10 µM hits and 10 µM only hits grouped by the target. (**E**) Enrichment of hit drug targets relative to their presence in the library is plotted versus the probability of random occurrence (p-value). A low p-value indicates that the probability of a target being enriched at random is low. Gray points indicate targets whose enrichment was not significant, colored points with labels denote significantly enriched targets (p<0.05). (**F**) Validation of selected hits from different target classes in multiple cell lines is shown. For each cell line, normality scores were normalized to their own DMSO controls. All drugs caused significant (p<0.05) reductions in normality scores in all cell lines.

The online version of this article includes the following source data and figure supplement(s) for figure 2:

**Source data 1.** Source data for *Figure 2C-F*.

**Figure supplement 1.** Validation of nucleolar stress hits and nucleolar morphology in different cell lines.

**Figure supplement 1—source data 1.** Source data for *Figure 2—figure supplement 1A and C*.

validated (*Figure 2—figure supplement 1A*). The complete list of hits is provided in *Supplementary file 1*.

All hits in the screen had normality scores lower than the control, that is, this parameter only went down, not up, in drug-treated cells. The number of cells in hit wells was typically lower than in control wells, indicating that the majority of drugs that induced nucleolar stress were cytostatic or cytotoxic (*Figure 2C*). The screening process did not distinguish whether the cytotoxic effects of the identified hits were a result of inhibiting their intended targets, impacting the nucleolus, or a combined effect. It is important to note that a low normality score is not necessarily a consequence of reduced viability because many drugs in the screen were cytostatic/cytotoxic without causing nucleolar stress. Rather, it underscores the fact that inhibition of nucleolar biological processes is overall detrimental to viability and proliferation. One of the internal positive controls for nucleolar stress in the screen was the compound BMH-21 – a well-characterized RNA Pol I inhibitor present in the library. BMH-21 intercalates in the DNA and binds strongly to GC-rich rDNA, repressing RNA Pol I transcription (*Colis et al., 2014*; *Wei et al., 2018*). BMH-21 induced a canonical nucleolar stress phenotype with dispersed nucleolin and segregation of UBF into stress caps. Cells treated with BMH-21 showed a 7.7-fold reduction in the normality score and a 2-fold reduction in cell number compared to DMSO control (highlighted in *Figure 2C*).

The anticancer compound library contained chemical inhibitors for various targets, mostly enzymes. Grouping hits by drug target showed that inhibitors of mTOR and PI3 kinase had the highest frequency among all hits. Other frequently hit drug targets were HSP90, Topoisomerases, and CDKs (*Figure 2D*). However, the overall representation of targets in the library varied: prioritized cancer targets and highly druggable targets were among the most represented. Since the representation of targets in the library was not equivalent, we calculated the enrichment of targets among hits relative to their presence in the library. The most significantly enriched targets (p<0.001) were HSP90, mTOR, PI3K, and topoisomerase inhibitors. Among other significantly enriched targets (p<0.05) were inhibitors of dihydrofolate reductase (DFHR), proteasome, CDKs, and other kinases (*Figure 2E*).

To ensure that the drug responses were not unique to RPE1 cells, validation was performed on additional cell lines with a panel of selected potent hits from different target classes: HSP90 inhibitors – 17-AAG, onalespib, BIIB-21; CDK inhibitors – dinaciclib, flavopiridol, LY2857785; proteasome inhibitors – carfilzomib and oprozomib; mTOR inhibitor sapanisertib; PI3K inhibitor taselicib; and topoisomerase inhibitors camptothecin and doxorubicin. This panel of drugs was validated in four other cell lines: two hTERT-immortalized cell lines – BJ5TA skin fibroblasts and CHON-002 fibroblasts, and two cancer-derived cell lines – DLD1 colon adenocarcinoma and HCT116 colon carcinoma. In all experimental cell lines, raw nucleolar normality scores before the drug treatments were different. Cancer cell lines had lower starting normality scores than hTERT cell lines (*Figure 2—figure supplement 1B and C*). To compensate for this initial difference, the results of drug treatments from each cell line were normalized to the vehicle control of that cell line. The degree of reduction in nucleolar normality scores varied between cell lines, which could be attributed to differences in baseline normality scores, as well as proteomic and metabolic shifts, alterations in signaling pathways that control ribosome production, and, potentially, variations in intracellular drug levels. Nonetheless, all compounds caused a significant reduction in nucleolar normality scores in all cell lines (*Figure 2F*). This result ensures that the nucleolar stress induced by these drugs was not specific to a particular cell line.

## Characterization of nucleolar stress induced by selected inhibitors

Canonical nucleolar stress induced by Pol I inhibitors is linked to reduced rRNA production. We measured the effect of the selected drug panel on rRNA synthesis by incorporation of 5-ethynyluridine (5-EU) into nascent RNA (*Jao and Salic, 2008*). Since ribosomal RNA can account for ~80% of the total cellular RNA (*Palazzo and Lee, 2015*), the total amount of nascent RNA approximates the synthesis of ribosomal RNA. All drugs in the panel caused a decrease in 5-EU incorporation, but to varying degrees. The level of reduction was similar within the same classes of drugs based on target, but different between classes (*Figure 3A*). Correlation analysis with normality scores showed that there was a trend for drugs with lower normality scores to have lower rRNA synthesis, but it was not statistically significant (*Figure 3B*). Furthermore, nucleolar stress phenotypes were distinct by target (*Figure 3—figure supplement 1*). This lack of significant correlation implied that the normality score may not be explained by a reduction in rDNA transcription alone.

Inhibitors of mTOR and PI3 kinase had the highest representation among all hits in the anticancer compound library. mTOR and PI3K are metabolic pathways that positively regulate ribosome biogenesis on multiple levels including rDNA transcription (*Mayer and Grummt, 2006*; *Pelletier et al., 2018*), so the strong (~60%) reduction in 5-EU incorporation in mTOR inhibitor sapanisertib and PI3K inhibitor taselicib was predictable. The reduction in normality score was likely a consequence of inhibiting upstream activating pathways that stimulate rDNA transcription and ribosome biogenesis.

Another major class of drugs that induced low normality scores were inhibitors of topoisomerase II, particularly anthracyclines that intercalate into DNA and act as topoisomerase poisons (doxorubicin, epirubicin, idarubicin, daunorubicin, pirarubicin, mitoxantrone, pixantrone). All DNA intercalating topoisomerase poison hits caused nucleolar shrinkage, rounding, and the canonical stress caps associated with RNA Pol I inhibition. Notably, actinomycin D and CX-5461 can also poison the action of topoisomerases (*Trask and Muller, 1988*; *Bruno et al., 2020*). Topoisomerase activity may be needed to resolve topological stress at the rDNA to continue transcription. rDNA transcription may be hypersensitive to DNA intercalators in general (*Andrews et al., 2021*), and for many DNA intercalating drugs the nucleolar stress cap phenotype is well-characterized (*Ferreira et al., 2020*).

From this point on, we further characterized the effects of representative drugs from non-intercalating, non-metabolic classes with less explored nucleolar stress phenotypes: HSP90 inhibitor 17-AAG (tanespimycin), proteasome inhibitor carfilzomib (kyprolis), and CDK inhibitor flavopiridol (alvocidib). 17-AAG blocks the ATP-binding pocket of molecular chaperone HSP90, leading to the accumulation of misfolded proteins (*Trepel et al., 2010*). Carfilzomib inhibits the chymotrypsin-like activity of the 20S proteasome, blocking the degradation of poly-ubiquitinated proteins (*Orlowski and Kuhn, 2008*). Flavopiridol is an ATP-competitive inhibitor of CDKs and can be considered a pan-CDK inhibitor (*Senderowicz, 1999*).

For detailed visualization of nucleolar morphology, we performed fluorescent in situ hybridization with antibody immunolabeling (immuno-FISH), where ribosomal DNA and nucleolar proteins UBF and nucleolin were labeled simultaneously (*Figure 3C*). In untreated cells, this labeling delineated the normal nucleolar anatomy: rDNA with bound UBF comprised the fibrillar center of the nucleolus, surrounded by granular component marked by nucleolin. Pol I inhibitor CX-5461, our positive control for nucleolar stress, induced classic peripheral stress caps with rDNA wrapped around condensed UBF foci (high-resolution images of rDNA and UBF are shown in magnified inserts in *Figure 3C*).

HSP90 inhibitor 17-AAG caused mild rDNA and UBF condensation that resembled the formation of stress caps but was less severe. It was recently shown that misfolded proteins can accumulate in the nucleolus upon heat shock (*Azkanaz et al., 2019*; *Frottin et al., 2019*). Given the ~40% reduction in rRNA synthesis, this stress phenotype may be brought about partly by reduced Pol I transcription and partly by the accumulation of misfolded proteins inside the nucleolar compartment.

The nucleolar stress phenotype induced by proteasome inhibitor carfilzomib was similar to that of 17-AAG, except that many nucleoli contained a diffuse pool of UBF not associated with the rDNA (*Figure 3C*, arrow). Proteasome inhibition was previously shown to cause nucleolar accumulation of polyubiquitinated proteins, termed aggresomes (*Latonen et al., 2011*). In addition, UBF itself can be ubiquitinated (*Liu et al., 2007*). We speculate that in addition to reduced transcription (by ~50%), this type of stress phenotype may reflect the nucleolar accumulation of polyubiquitinated proteins normally targeted for degradation. As with 17-AAG stress, rRNA production may be directly impaired by abnormal protein accumulation in the nucleolus, or these two factors may be linked indirectly.

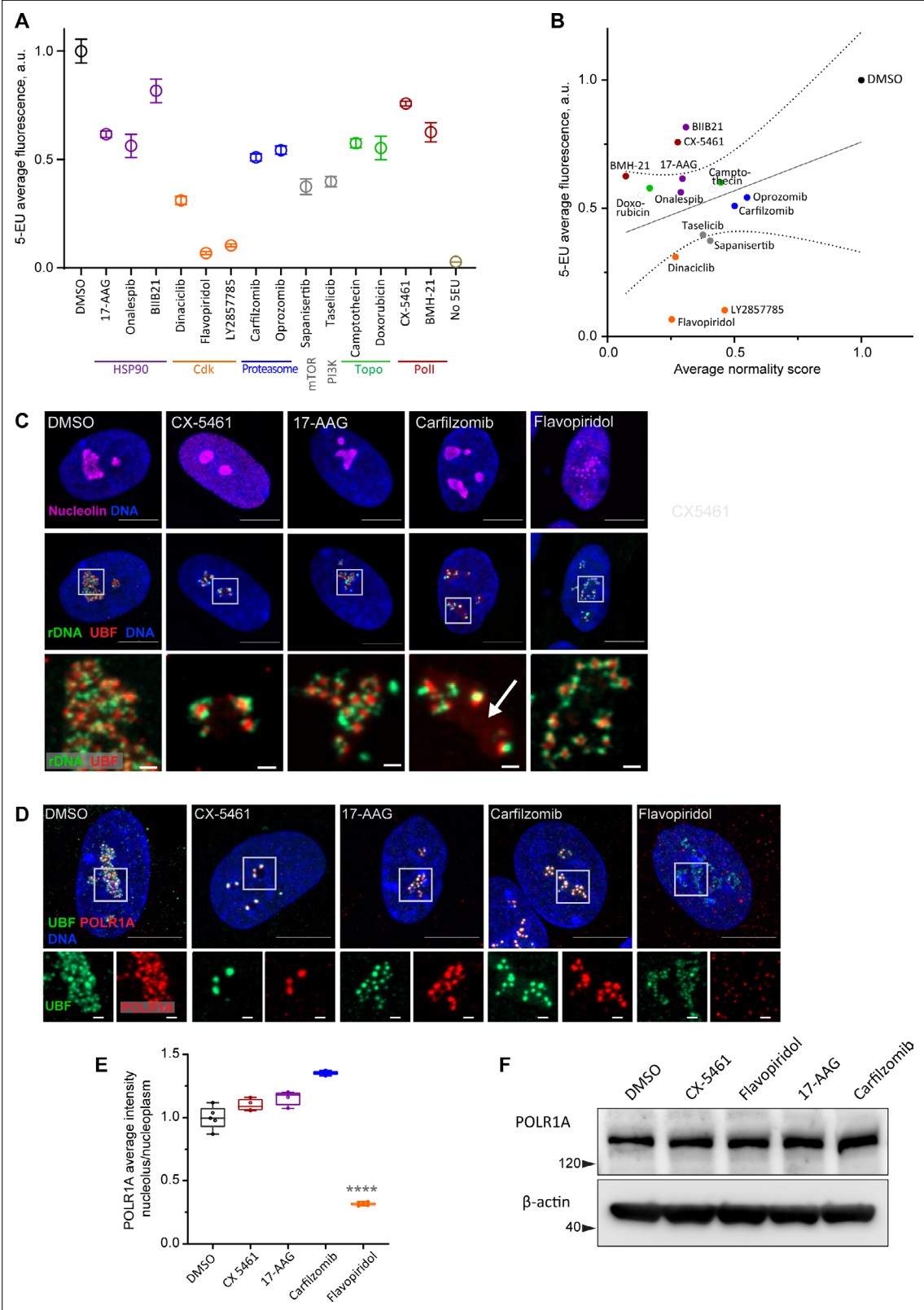

**Figure 3.** Characterization of nucleolar stresses in a panel of selected drugs. (**A**) 5-ethynyluridine (5-EU) incorporation was measured in RPE1 cells treated with the panel of selected drug hits from the screen. All compounds were added for 10 hr followed by 4 hr of 0.5 mM 5-EU incorporation. All drugs were at 10 μM concentration except LY2857785 and CX-5461 were used at 2.5 μM, camptothecin and flavopiridol were used at 5 μM, and doxorubicin and BMH-21 at 1 μM. 5-EU-labeled RNA was detected with fluorescent azide and quantified by imaging. Plots represent means with

*Figure 3 continued on next page*

*Figure 3 continued*

standard deviations of three or more large fields of view containing hundreds of cells. Raw fluorescent intensity values were normalized to the average of the DMSO controls. All drug treatments caused a significant reduction in 5-EU incorporation compared to DMSO (p<0.01, unpaired *t*-tests). (**B**) A correlation plot of average nucleolar normality scores versus average 5-EU fluorescence is shown. Both parameters were normalized to the average of the DMSO controls. The trend for drugs with lower normality scores to have lower 5-EU incorporation was not significant (Pearson's *r* = 0.33, p=0.23). (**C**) Fluorescent in situ hybridization with antibody immunolabeling (immuno-FISH) images of drug-treated RPE1 cells labeled with human rDNA probe (green), UBF (red), and nucleolin (magenta) are shown. Nuclei were counterstained with DAPI (blue). Bar, 10 µm. The duration of 2.5 µM CX-5461 and 10 µM flavopiridol treatments was 5 hr, 10 µM 17-AAG and 10 µM carfilzomib 10 hr. Magnified inserts show details of individual nucleoli (bar, 1 µm). Note peripheral stress caps in CX-5461 and unfolded rDNA/UBF in flavopiridol–treated cells. The arrow in the carfilzomib panel indicates the diffuse pool of UBF not associated with rDNA. (**D**) Immunofluorescence images of RPE1 cells treated as in (**C**) and labeled with antibodies against UBF (green) and POLR1A (red, antibody C-1). Nuclei were counterstained with DAPI. Bar, 10 µm. Magnified insets show details of individual nucleoli (bar, 1 µm). UBF and POLR1A label the same structures in all treatments except flavopiridol. (**E**) The quantification of POLR1A immunofluorescence from (**D**) is plotted. The box plot depicts ratios of POLR1A signal intensity in the nucleolus versus nucleoplasm normalized to the average of DMSO controls. The plot represents the means of 4–5 fields of view containing a total of 80–100 cells. Asterisks indicate a significant reduction in nucleolar POLR1A (p<0.0001, unpaired *t*-test flavopiridol vs. DMSO). (**F**) Western blot analysis of POLR1A protein levels in RPE1 cells treated with the indicated drugs for 8 hr. Total POLR1A levels were not altered.

The online version of this article includes the following source data and figure supplement(s) for figure 3:

**Source data 1.** Source data for *Figure 3A,B and E*.

**Source data 2.** Source data for *Figure 3F*.

**Figure supplement 1.** Examples of immunofluorescence labeling of RPE1 cells treated with a panel of indicated compounds.

**Figure supplement 2.** Nucleolar morphology changes after flavopiridol addition and washout, and effects of selected drugs on POLR1A localization.

**Figure supplement 2—source data 1.** Source data for *Figure 3—figure supplement 2D*.

The effect of CDK inhibitor flavopiridol on the nucleolus was profound and entirely different from all other stress phenotypes. Normally, rDNA and UBF form a compacted structural scaffold of the nucleolus. In CDK inhibitor-treated cells, this scaffold extended into undulant fibers, while nucleolin fully dispersed in the nucleoplasm. Discernable nucleolar boundaries demarcated by granular component proteins were completely lost, implying that the nucleolar compartment disintegrated, and only the bare scaffold remained (*Figure 3C*, last panel).

We further investigated the effects of flavopiridol on nucleoli by live imaging of RPE1 cell lines stably expressing eGFP-nucleolin or eGFP-UBF. Within 2–3 hr after flavopiridol addition, eGFP-nucleolin became dispersed in the nucleoplasm forming many small round droplets within the diffused pool (*Figure 3—figure supplement 2A* and *Video 3*). In cells expressing eGFP-UBF, flavopiridol treatment induced UBF decompaction into dotted strings with some diffusion into the nucleoplasm (*Figure 3—figure supplement 2B* and *Video 4*). This effect was not consistent with apoptosis or necrosis, and it was fully reversible within 4–6 hr after the inhibitor was washed out (*Figure 3—figure supplement 2C* and *Video 5*).

Given that flavopiridol caused the most severe reduction in 5-EU incorporation (more than 80%), we measured the Pol I occupancy on rDNA by

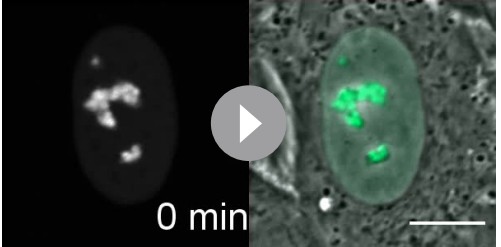

**Video 3.** Fluorescence and phase-contrast time-lapse video of a human RPE1 cell stably expressing eGFP-nucleolin that was treated with 10 µM CDK inhibitor flavopiridol. Nucleolin largely disperses forming small round droplets within the diffused nucleoplasmic pool. Time is indicated as minutes after drug addition. Bar, 10 µm.

https://elifesciences.org/articles/88799/figures#video3

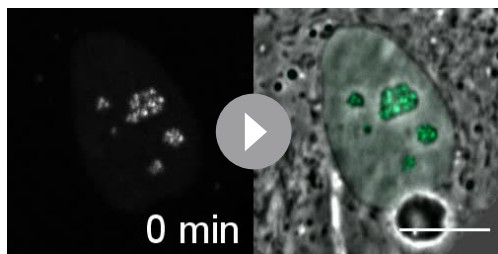

**Video 4.** Fluorescence and phase-contrast time-lapse video of a human RPE1 cell stably expressing eGFP-UBF that was treated with 10 µM CDK inhibitor flavopiridol. UBF de-compacts into strings with some diffusion into the nucleoplasm. Time is indicated as minutes after drug addition. Bar, 10 µm.

https://elifesciences.org/articles/88799/figures#video4

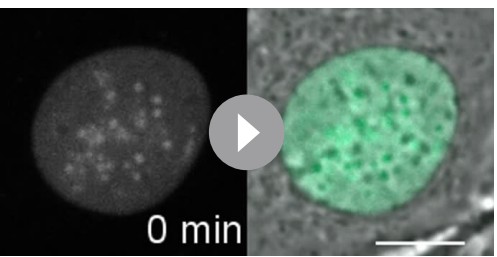

**Video 5.** Fluorescence and phase-contrast time-lapse video of a human RPE1 cell stably expressing eGFP-nucleolin after flavopiridol washout. The cell was pretreated with 10 µM flavopiridol for 5 hr, then the drug was washed out before imaging. Nucleoli re-form within 4–6 hr. Time is indicated as minutes after drug washout and initiation of imaging. Bar, 10 µm.

https://elifesciences.org/articles/88799/figures#video5

immunolabeling its catalytic subunit POLR1A. In the DMSO control and all drug treatments except flavopiridol, POLR1A was associated with the rDNA marked by UBF. In flavopiridol-treated cells, this association was reduced by nearly 70% (*Figure 3D and E*, *Figure 3—figure supplement 2D*). The total cellular amount of POLR1A protein did not decrease with any of the drug treatments (*Figure 3F*), indicating that the loss of POLR1A association with rDNA in flavopiridol was not due to its degradation. These data suggest that inhibiting CDK activity with flavopiridol creates a unique and extreme nucleolar stress state with only the bare scaffold remaining. This stress is associated with very low RNA Pol I transcription and disassociation of POLR1A from the rDNA.

## Inhibition of transcriptional CDK9 causes disassociation of RNA Pol I from rDNA and disintegration of the granular component of the nucleolus

Our subsequent experiments focused on discerning the mechanism of nucleolar stress induced by CDK inhibitors. The focus on CDK inhibitors was prompted by the severe and unexplained 'bare scaffold' phenotype they induced, implying the existence of an unknown mechanism through which CDKs regulate nucleolar structure and function. Furthermore, chemotherapeutic approaches involving CDK inhibitors demonstrated a high rate of clinical failure linked to their difficult-to-explain toxicity (*Whittaker et al., 2017*), which may be attributed at least in part to their unrecognized impact on nucleolar organization and ribosome biogenesis.

We speculated that the CDK activity was needed for maintaining RNA Pol I transcription. CDKs are serine-threonine kinases that require an activation subunit – a cyclin – to phosphorylate their substrates. The human genome encodes 21 CDKs, some of which have been studied extensively while others remain cryptic. There are well-studied CDKs that drive cell cycle progression (e.g., CDK1, CDK2, CDK4, CDK6). Transcriptional CDKs drive the activity of RNA polymerase II (CDK8, CDK9, CDK12, CDK19). There are also CDKs with poorly understood biological functions (CDK15, CDK18, CDK20) (*Malumbres et al., 2009*). For RNA Pol II transcription, CDK9 is the important kinase that phosphorylates the C-terminal domain (CTD) to control transcription initiation, elongation, and termination (*Bacon and D'Orso, 2019*). RNA Pol I lacks a CTD, and currently there are no known CDK-related mechanisms that control RNA Pol I.

Flavopiridol is a pan-CDK inhibitor. To pinpoint the specific CDK responsible for the observed phenotype, we tested three more CDK inhibitors that reportedly have selectivity for the transcriptional CDK9 and were not in our library: AZD4573 (*Barlaam et al., 2020*; *Cidado et al., 2020*), JSH-150 (*Wang et al., 2018*), and MC180295 (*Zhang et al., 2018*). In addition, we included the established catalytic inhibitor of RNA Pol II, α-amanitin (*Bushnell et al., 2002*; *Brueckner and Cramer, 2008*). Cells treated with AZD4573, JSH-150, and MC180295 showed an extended rDNA/UBF scaffold and dispersed nucleolin analogous to flavopiridol-treated cells (*Figure 4A*), suggesting that the nucleolar stress was likely induced by inhibition of CDK9. Quantification of nucleolar normality scores gave similar values (70–75% reduction) for all three CDK inhibitors (*Figure 4B*). Importantly, α-amanitin did not mimic these effects and only caused a minor reduction in the normality score, indicating that catalytic inhibition of RNA Pol II alone was insufficient to cause the nucleolar stress phenotype observed with pan-CDK and CDK9-specific inhibitors.

Production of nascent RNA, most of which is rRNA, was decreased by 70–80% in all CDK inhibitors as measured by incorporation of 5-EU (*Figure 4C*). The RNA Pol II inhibitor α-amanitin induced a small (~10%) but significant decrease in 5-EU incorporation, which was expected because 5-EU is also incorporated in RNA Pol II transcripts. To examine the amount of rRNA by another method, we measured the total amount of rRNA by Y10b (anti-rRNA) antibody fluorescence. Y10b antibody labeling showed

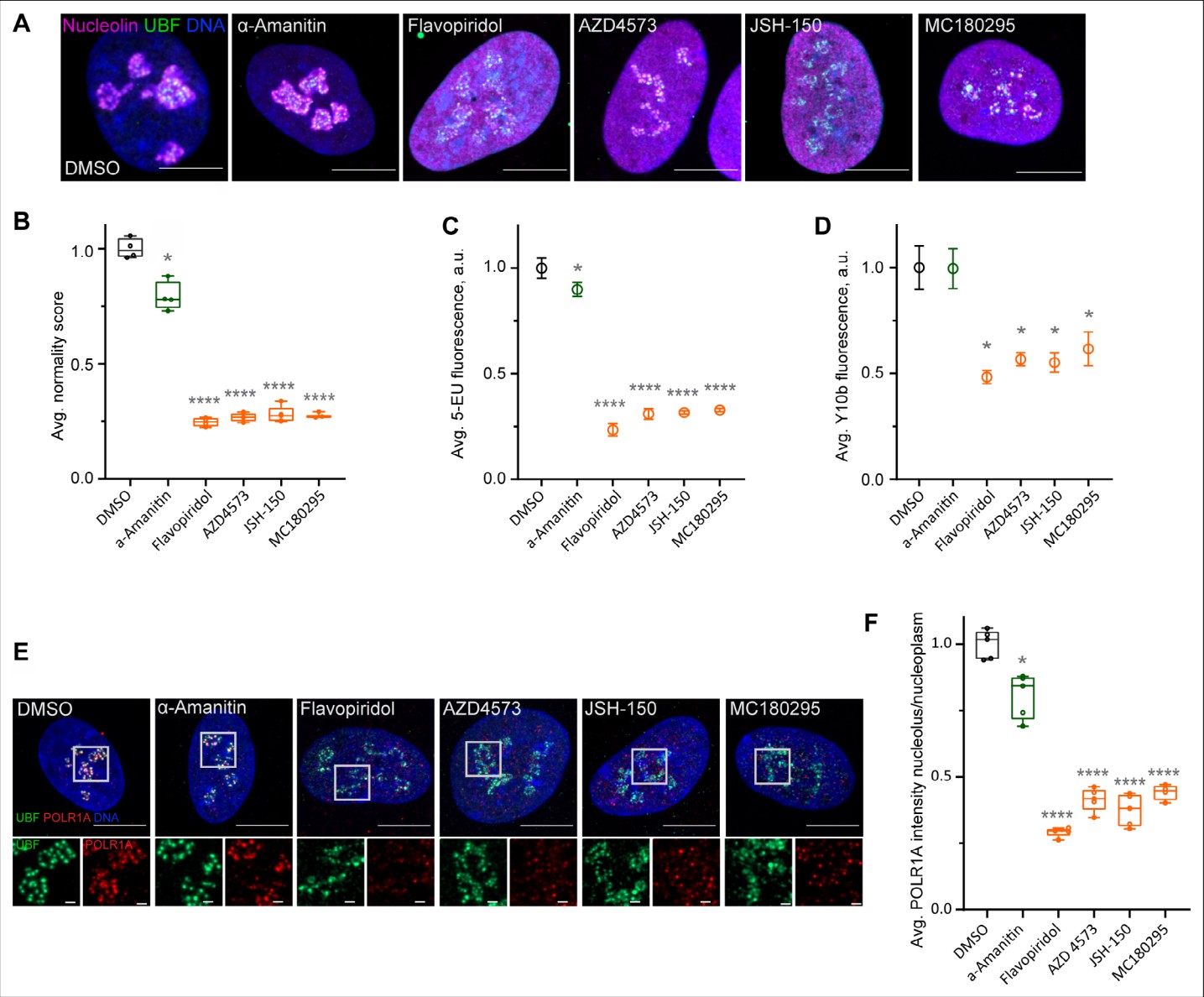

**Figure 4.** Effects of CDK9 inhibitors on nuclear morphology and function. (**A**) Immunofluorescence images of drug-treated cells labeled with antibodies against nucleolin (magenta) and UBF (green) are shown. Nuclei were counter-stained with DAPI. Cells were treated with DMSO (vehicle), 5 µg/ml Pol II inhibitor α-amanitin, 5 µM pan-CDK inhibitor flavopiridol, or 5 µM CDK9 inhibitors AZD4573, JSH-150, and MC180295 for 5 hr. Note that all CDK inhibitor-treated cells have similar phenotypes. Bar, 10 µm. (**B**) Normality score measurements of RPE1 cells treated as in (**A**) are shown. Raw normality scores were normalized to the average of DMSO treated cells. Box plots represent means of 4–5 fields of view containing many cells. Asterisks: *p<0.01, ****p<0.0001 (unpaired *t*-test comparing treatments vs. DMSO). (**C**) 5-Ethynyluridine (5-EU) incorporation in RPE1 cells treated as in (**A**) is shown. 5-EU-labeled RNA was detected with fluorescent azide and quantified by imaging. Plots represent means with standard deviations of 5–8 large fields of view containing hundreds of cells. Raw fluorescence intensity values were normalized to the average of the DMSO-treated control cells. Asterisks: *p<0.01, ****p<0.0001 (unpaired *t*-test comparing treatment vs. DMSO). (**D**) Quantification of Y10b (anti-rRNA) antibody labeling of RPE1 cells treated as in (**A**). Plots represent means with standard deviations of three large fields of view containing hundreds of cells. Raw fluorescent intensity values were normalized to the average of the DMSO control. Asterisks indicate p<0.01 (unpaired *t*-test comparing treatments vs. DMSO). (**E**) Immunofluorescence images of RPE1 cells treated as in (**A**) and labeled with antibodies against UBF (green) and POLR1A (red) are shown. Nuclei were counter-stained with DAPI. Bar, 10 µm. Magnified inserts show details of individual nucleoli (bar, 1 µm). (**F**) Quantification of POLR1A immunofluorescence from (**E**) is shown. The box plot depicts ratios of POLR1A signal intensity in the nucleolus versus nucleoplasm normalized to the average of DMSO-treated cells. Plots represent means of 4–5 fields of view containing many cells. Asterisks: *p<0.01, ****p<0.0001 (unpaired *t*-test comparing treatment vs. DMSO).

The online version of this article includes the following source data for figure 4:

**Source data 1.** Source data for *Figure 4B,C,D,F*.

45–55% decrease in total rRNA that was comparable in all CDK inhibitors. α-Amanitin treatment did not cause a significant decrease in total rRNA (*Figure 4D*). These data showed that CDK inhibition caused a dramatic reduction in RNA Pol I function.

Next, we measured RNA Pol I occupancy on the rDNA by immunofluorescence labeling of POLR1A and rDNA marker UBF. In flavopiridol-treated cells, POLR1A association with rDNA was reduced by ~70%, and in CDK9 inhibitors it was reduced by ~60% (*Figure 4E and F*). The effect of α-amanitin on the association of POLR1A with rDNA was much smaller (20%). These data suggest that CDK inhibition reduces rRNA production by causing the disassociation of RNA Pol I from the rDNA, and not through a secondary effect of inhibition of RNA Pol II. Our data further suggest that transcriptional CDK activity, potentially CDK9, is necessary for RNA Pol I activity and nucleolar integrity.

## Multiple nucleolar proteins are phosphorylated by CDK9, including Treacle, the transcriptional co-activator of RNA Pol I

CDKs phosphorylate many proteins, yet the scope of their targets in the interphase nucleolus is largely unexplored. To search for potential CDK target proteins in the nucleolus, we used mass spectrometry combined with titanium dioxide phosphopeptide enrichment. Nuclear lysates were made from untreated RPE1 cells and cells treated with pan-CDK inhibitor flavopiridol or CDK9-specific inhibitor AZD4573. Tryptic peptides were then prepared from equal amounts of each lysate. Also, 10% of each peptide sample was used to measure the total protein abundance by MudPIT, and 90% of each sample was enriched for phosphopeptides followed by Orbitrap-based mass spectrometry analysis (*Figure 5A*).

The total protein abundance analysis identified 61 enriched and 22 depleted proteins in both drug treatments (*Figure 2B*). Enrichment and depletion from nuclear extracts could be due to both changes in synthesis/degradation or nuclear import/export rates. Among commonly enriched proteins with the highest fold change was the stress-induced transcription factor p53. The tumor suppressor protein p53 has previously been shown to accumulate in flavopiridol-treated cells by multiple studies (*Shapiro et al., 1999*; *Alonso et al., 2003*; *Demidenko and Blagosklonny, 2004*), confirming our quantitative proteomics results. The commonly depleted group of proteins included ribosome biogenesis factors RPF1, RRP36, and DDX56. Nuclear export or degradation of these factors could occur due to nucleolar disassembly caused by CDK inhibition. The complete list of enriched and depleted proteins is provided in *Supplementary file 2*.

The subsequent phosphoproteomics approach was focused on identifying proteins that became dephosphorylated in cells treated with CDK inhibitors using titanium dioxide phosphopeptide enrichment. The number of spectra for phosphorylated peptides recovered from untreated samples was compared to treated samples. Peptides from proteins that were significantly depleted by drug treatments were excluded from this analysis. We detected 148 proteins with peptides that had lower phosphorylation in both treatments (*Figure 5C*, *Supplementary file 3*). Most of the phosphorylation sites were serines and threonines. Multiple identified phosphosites belonged to POLR2A, the catalytic subunit of RNA polymerase II (Pol II). Transcriptional CDKs are recognized for their prominent role in regulating the activity of Pol II by phosphorylating the unique C-terminal domain (CTD) in POLR2A, which is absent in POLR1A (*Burton, 2014*; *Parua and Fisher, 2020*; *Barba-Aliaga et al., 2021*). In particular, CDK9 phosphorylates the CTD on multiple residues, controlling transcription initiation, elongation, and recruitment of the splicing machinery (*Eick and Geyer, 2013*; *Guo et al., 2019*). We recovered differentially phosphorylated POLR2A peptides for eight C-terminal residues (*Figure 5C*, red rings), validating this approach for detecting CDK substrates.

Out of 148 proteins with lower phosphorylation after treatment with CDK inhibitors, 27 were nucleolar proteins (*Figure 5C*, denoted by yellow circles). The list of nucleolar CDK targets included proteins involved in multiple steps of ribosome biogenesis, including rRNA processing, assembly of ribosomal subunits, as well as architectural nucleolar proteins. Lower phosphorylation of these components could change the function and/or affinity of these proteins, contributing to the nucleolar disassembly phenotype. For instance, the Ki-67 protein implicated in organizing heterochromatin around the nucleolus (*Sobecki et al., 2016*) had lower phosphorylation on four sites and became speckled throughout the nucleus (*Figure 5—figure supplement 1A*). Given that POLR1A disassociates from the rDNA in CDK inhibitors, we searched for a substrate that could provide a link for Pol I association with the rDNA. Key rDNA transcription factors UBF and RRN3, as well as the components

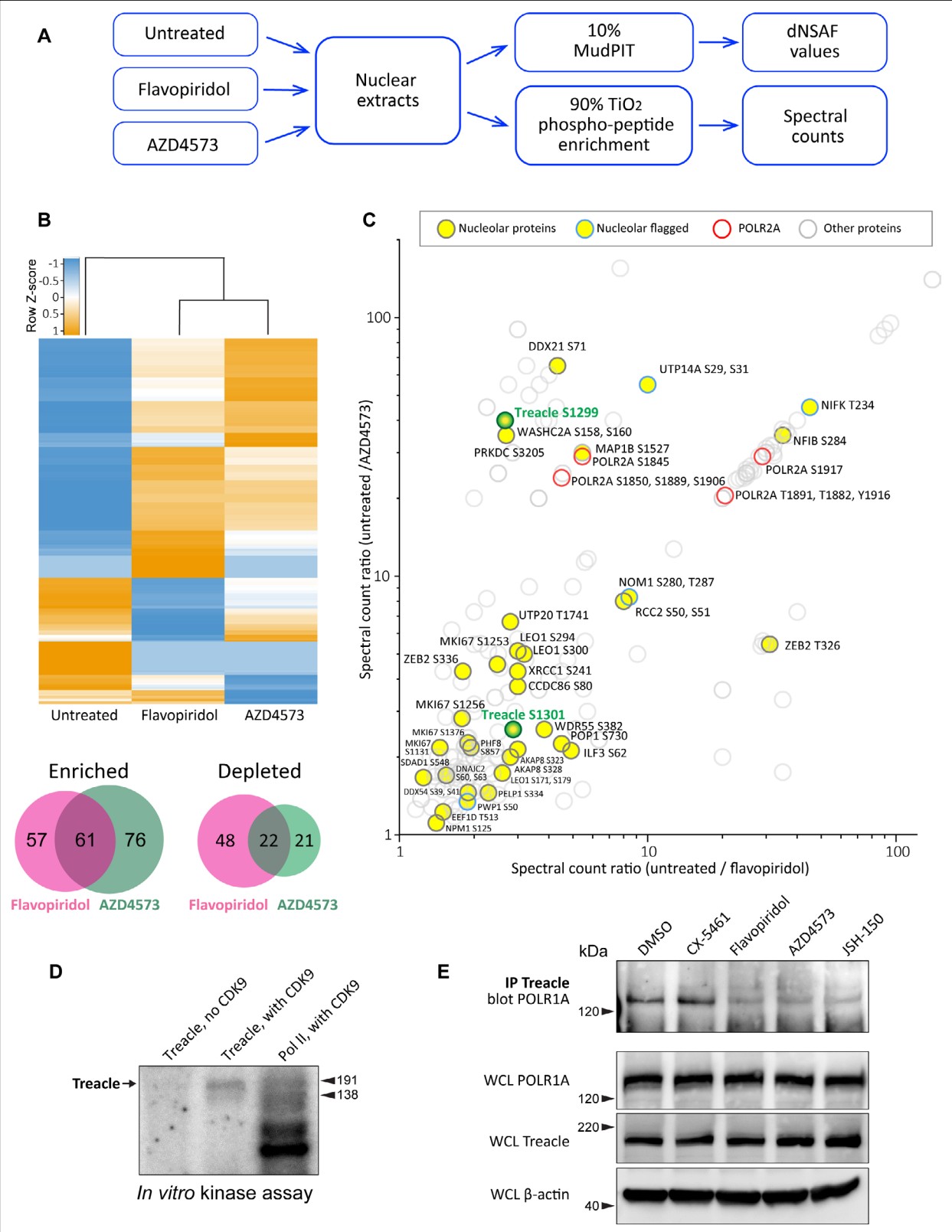

**Figure 5.** Effects of CDK9 inhibitors on the nuclear proteome and phosphoproteome. (**A**) An overview of the nuclear phosphoprotein profiling workflow is shown. Nuclear protein extracts were generated from RPE1 cells or following treatment with 5 μM flavopiridol or 5 μM AZD4573 for 5 hr. After trypsin digestion, peptide samples were split into two parts: 10% of each sample was used for MudPIT and 90% was enriched for phosphopeptides using High-Select TiO₂ phosphopeptide enrichment spin tips (Thermo Scientific). Peptides prepared from nuclear extracts and phosphopeptide-enriched

*Figure 5 continued on next page*

*Figure 5 continued*

samples were analyzed using an Orbitrap Elite Mass Spectrometer coupled with an Agilent 1260 Infinity quaternary pump (two biological replicates per condition). (**B**) Nuclear proteins selectively enriched or depleted in cells treated with indicated CDK inhibitors were determined. The clustered heatmap shows Z-score values (calculated from dNSAF values) for proteins enriched or depleted in drug-treated cells (LogFC> ± 1.5, QPROT FDR < 0.05). The Venn diagrams depict overlaps between enriched and depleted proteins in both drug treatments. The full list of proteins is provided in *Supplementary file 2*. (**C**) Phosphopeptides that decreased after drug treatment are depicted. Axes indicate the ratio of numbers of spectra for phosphopeptides detected in untreated relative to drug-treated samples. If no phosphopeptides were detected in drug-treated samples, the number 0 was replaced with 0.1 to avoid dividing by zero. Proteins with significantly decreased abundance after drug treatment (QPROT FDR <0.0001) were filtered out. In addition, phosphosites with fewer than two spectral counts in either of the untreated samples were not considered. Yellow circles indicate proteins with the GO-term nucleolus for the subcellular compartment, labeled with phosphorylation sites. Blue outlines flag proteins with non-significant dNSAF decrease in any of the treatments. Red outlines denote phosphopeptides belonging to POLR2A, labeled with sites. All other nuclear phosphopeptides are shown as open circles. Treacle phosphosite labels are highlighted in green. The full list of phosphopeptides with sites is provided in *Supplementary file 3*. (**D**) Representative in vitro CDK9/cyclin K kinase assay with recombinant human Treacle protein substrate and radiolabeled [γ-$^{32}$P] ATP is shown. The negative control (first lane) contained the Treacle substrate without the kinase. The second lane contained both the kinase and the substrate. In the third lane, the Pol II holoenzyme complex purified from *S. cerevisiae* served as a positive control. Molecular weights of RNA Pol II holoenzyme subunits Rbp1 (191 kDa) and Rbp2 (138 kDa) are indicated. (**E**) Immunoprecipitation (IP) of Treacle from drug-treated cells followed by western blotting with anti-POLR1A antibody. Treacle antibody pulled down POLR1A from lysates of cells treated with DMSO or CX-5461, but not from cells treated with CDK inhibitors. Lower panels show POLR1A and Treacle immunoblots of corresponding whole-cell lysates (WCL). β-Actin was used as a loading control.

The online version of this article includes the following source data and figure supplement(s) for figure 5:

**Source data 1.** Source data for *Figure 5D*.

**Source data 2.** Source data for *Figure 5E*.

**Figure supplement 1.** Effects of selected drugs on localization of Ki-67 and Treacle, and additional in vitro CDK9 kinase assays.

**Figure supplement 1—source data 1.** Source data for *Figure 5—figure supplement 1B*.

**Figure supplement 1—source data 2.** Source data for *Figure 5—figure supplement 1C*.

of selectivity factor 1 (SL1) that promote RNA Pol I transcription initiation (*Friedrich et al., 2005*) and POLR1A itself, were absent from our list.

One notable candidate for linking POLR1A to the rDNA was the protein Treacle, encoded by the *TCOF1* gene. Mutations in *TCOF1* cause Treacher Collins syndrome, a congenital craniofacial disorder associated with tissue-specific disruptions in ribosome biogenesis (*Sakai and Trainor, 2009*; *Noack Watt et al., 2016*). Treacle is a large nucleolar protein containing multiple low-complexity regions with alternating acidic and basic tracts in its central disordered region (*Grzanka and Piekiełko-Witkowska, 2021*). Treacle was shown to be involved in rDNA transcription by connecting UBF and Pol I (*Valdez et al., 2004*), and was also reported to recruit Pol I machinery independently of UBF (*Lin and Yeh, 2009*). Treacle is a phosphoprotein that contains numerous serine and threonine residues within its central disordered domain, but the functional significance of this phosphorylation is unknown.

The phosphorylation of Treacle decreased when cells were treated with CDK inhibitors. However, this could be attributed to direct or indirect effects of inhibiting CDK9. To test if CDK9 can phosphorylate Treacle directly, we performed a radioactive in vitro kinase assay with recombinant CDK9/cyclin K complex using recombinant Treacle protein as a substrate. CDK9/cyclin K effectively phosphorylated recombinant Treacle in vitro (*Figure 5D*). Purified RNA Pol II holoenzyme complex was used as a positive control, showing multiple phosphorylated bands as expected. Treacle kinase assays using recombinant CDK9 complexed with cyclin T were not successful presumably due to the low activity of this recombinant protein complex, as indicated by minimal phosphorylation of a known substrate, RNA Pol II (*Figure 5—figure supplement 1B*). In line with the phosphoproteomics analysis, CDK9 did not phosphorylate purified RNA Pol I holoenzyme (*Figure 5—figure supplement 1C*). Therefore, CDK9 can directly phosphorylate Treacle and RNA Pol II, but not RNA Pol I.

Treacle localized to the fibrillar center of the nucleolus together with UBF in untreated cells. This localization was unaffected in cells treated with CDK inhibitors (*Figure 5—figure supplement 1D*). Therefore, CDK inhibition did not impact its subcellular localization as it did for POLR1A. Next, we asked if CDK inhibition affected Treacle interaction with POLR1A. For this, we immunoprecipitated Treacle protein from untreated and drug-treated cell lysates and probed for POLR1A. Anti-Treacle antibody efficiently pulled down POLR1A from lysates of cells treated with vehicle or Pol I inhibitor CX-5461, but not from cells treated with pan-CDK inhibitor flavopiridol or CDK9-specific inhibitors AZD4573 and JSH-150 (*Figure 5E*). Western blotting of whole-cell lysates did not show the

degradation of Treacle or POLR1A in any drug treatments. These results suggest that the phosphorylation of Treacle, possibly by CDK9, plays a significant role in recruiting Pol I machinery to the rDNA to facilitate Pol I transcription.

In summary, inhibition of CDK9 creates an extreme form of nucleolar stress where only the bare rDNA scaffold with few associated proteins remains. The subunits of the Pol I machinery dissociate from rDNA, potentially due in part to the dephosphorylation of Treacle. Thus, transcriptional CDK activity must be necessary for maintaining rDNA transcription and nucleolar integrity.

## Biophysical properties of nucleoli under RNA Pol I versus CDK inhibition

To better understand how the transcriptional state impacts nucleolar integrity, we investigated biophysical properties of nucleoli by probing the molecular dynamics of NPM1 (nucleophosmin) using FRAP. NPM1 is a multifunctional nucleolar protein involved in the assembly of ribosomal subunits that occupies the granular component (GC) of the nucleolus. It contains an N-terminal oligomerization domain, a C-terminal RNA-binding domain, an intrinsically disordered region (IDR), and multiple acidic tracts throughout the protein. It forms a homopentamer, binds rRNA, and can form multivalent interactions with other proteins that contain arginine-rich domains. Studies of IDR-containing proteins including NPM1 demonstrated that these proteins can form homotypic and heterotypic interactions that drive liquid–liquid phase separation (LLPS), which has been proposed to play a role in the assembly of membrane-less organelles such as the nucleolus (*Feric et al., 2016*; *Mitrea et al., 2018*; *Lafontaine et al., 2021*). We generated an RPE1 cell line in which the endogenous NPM1 is monoallelically tagged with monomeric eGFP and used it to study the molecular exchange within nucleoli and between nucleoli and nucleoplasm.

First, we determined the dynamics of NPM1 in normal, untreated RPE1 cells. For this, we performed FRAP analysis of NPM1-GFP by photobleaching a whole nucleolus or a part of the nucleolus. For a classical phase-separated liquid condensate, the recovery time of a partially bleached structure was expected to be faster than the recovery of the entire structure due to rapid internal molecular rearrangements (*Brangwynne et al., 2009*). There was no significant difference in the recovery rate between fully bleached and partially bleached nucleoli (*Figure 6A and B*). Full- and half-FRAP regions were roughly the same size. The average recovery rate ($T_{1/2}$) was ~28 ± 9.9 s for fully bleached nucleoli and ~24 ± 7.8 s for partially bleached nucleoli. This similarity in the $T_{1/2}$ was also true for cells treated with RNA Pol I inhibitor CX-5461 (~11 ± 3.4 s and ~10 ± 3.2 s). Nucleoplasmic recovery rates were much faster than nucleolar recovery rates in both cases. In cells with more than one nucleolus, the partial bleach analysis revealed that as the photobleached part gained and the unbleached part expended fluorescence, the separate, unconnected nucleoli also lost fluorescence at a comparable rate (an example is shown in *Figure 6C*). This implied that nucleolar and nucleoplasmic pools of NPM1 both contributed to the recovery process,

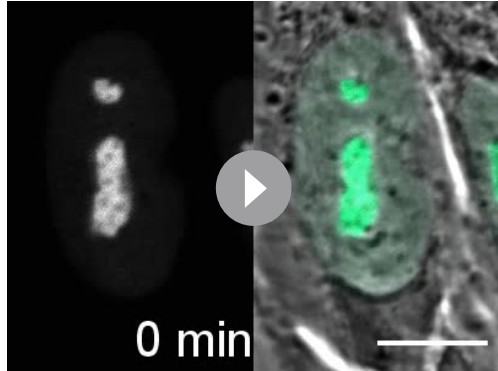

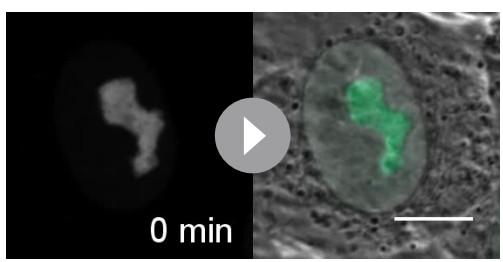

**Video 6.** Fluorescence and phase-contrast time-lapse video of a human RPE1 cell with endogenous NPM1 tagged with eGFP treated with 2.5 µM CX-5461. NPM1-GFP concentrates inside the nucleolar remnants over time. Time is indicated as minutes after drug addition. Bar, 10 µm.
https://elifesciences.org/articles/88799/figures#video6

**Video 7.** Fluorescence and phase-contrast time-lapse video of a human RPE1 cell with endogenous NPM1 tagged with eGFP treated with 10 µM flavopiridol. NPM1-GFP disperses into multiple small round globules, and the fluorescent intensity of the nucleoplasm pool increases. Time is indicated as minutes after drug addition. Bar, 10 µm.
https://elifesciences.org/articles/88799/figures#video7

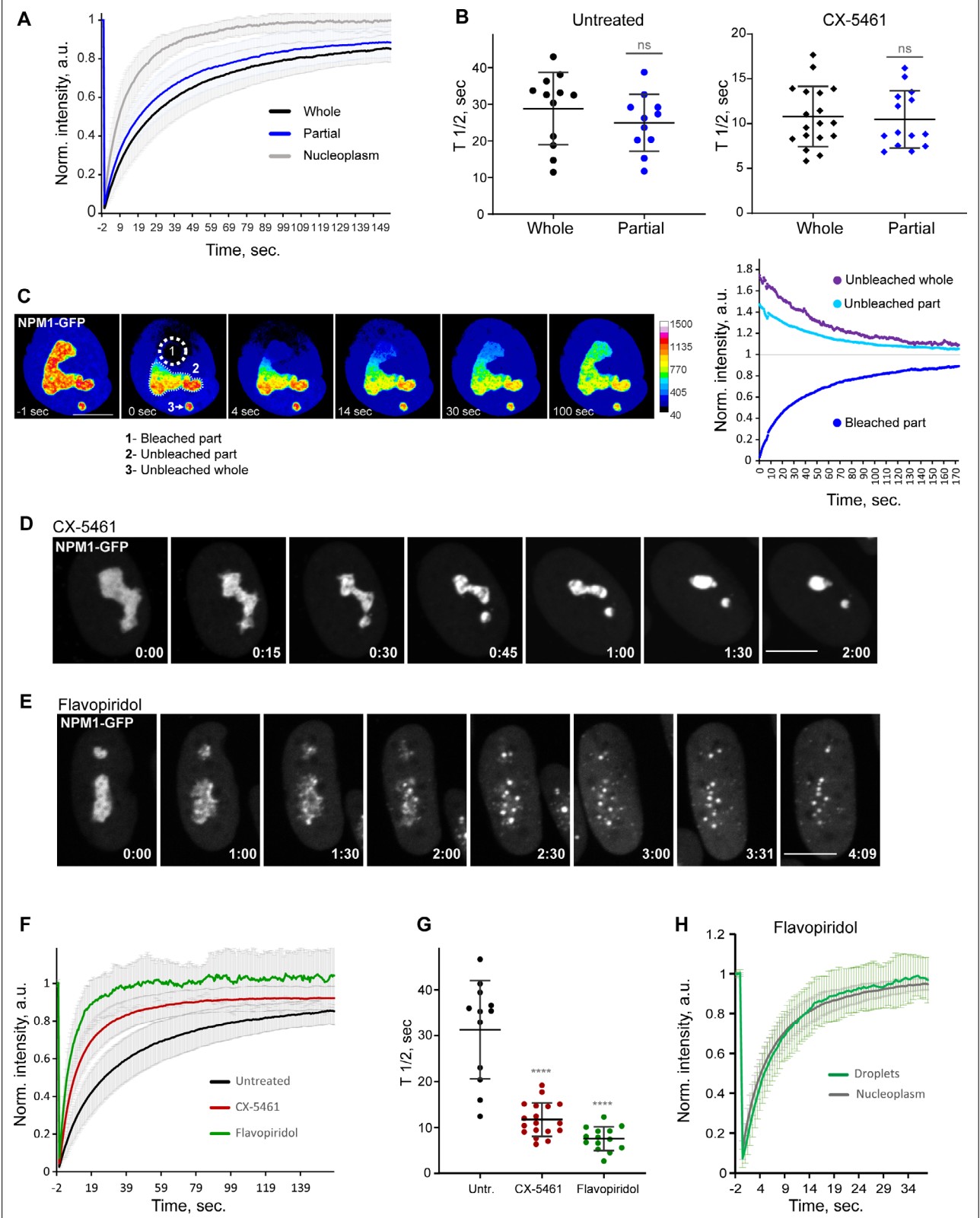

**Figure 6.** Analysis of biophysical properties of the nucleoli using endogenous NPM1 tagged with GFP. (**A**) Fluorescence recovery after photobleaching (FRAP) analysis of NPM1-GFP in whole-bleached or partially bleached nucleoli in untreated cells is shown. Plots are averages of normalized fluorescence intensities of 10 cells or more. Bars denote standard deviation. (**B**) Individual $T_{1/2}$ measurements of whole and partially bleached nucleoli are shown in untreated cells and in cells treated with 2.5 µM Pol I inhibitor CX-5461. The difference between whole and partially bleached nucleoli was not significant

*Figure 6 continued on next page*

*Figure 6 continued*

in both groups (t-test). (**C**) An example of a partially bleached nucleolus in a cell with two nucleoli is shown. The normalized intensity plot on the left shows the fluorescence recovery of the photobleached part (1), the corresponding loss of fluorescence in the unbleached part (2), and the loss of fluorescence in the separate unbleached nucleolus (3). Bar, 10 μm. (**D**) Time-lapse images of NPM1-GFP expressing cell treated with 2.5 μM CX-5461 at time 0 are shown. NPM1-GFP concentrated inside the nucleolar remnants over time. The complete video sequence is shown in *Video 6*. Bar, 10 μm. (**E**) Time-lapse images of NPM1-GFP expressing cell treated with 10 μM flavopiridol at time 0 are shown. NPM1-GFP dispersed into multiple small round globules, and the fluorescent intensity of the nucleoplasm pool increased. The complete video sequence is shown in *Video 7*. Bar, 10 μm. (**F**) FRAP analysis of NPM1-GFP in untreated cells (whole nucleoli) and cells treated with 2.5 μM CX-5461 (whole remnants) or 10 μM flavopiridol (whole globules) is shown. Plots are averages of normalized fluorescence intensities of more than 10 cells. Bars denote standard deviation. (**G**) Corresponding individual $T_{1/2}$ measurements in untreated cells and cells treated with 2.5 μM CX-5461 or 10 μM flavopiridol are shown. Asterisks indicate p<0.0001 (unpaired t-test comparing treated groups to untreated). (**H**) FRAP analysis of NPM1-GFP in cells treated with flavopiridol is compared for droplets versus nucleoplasm. Plots are averages of normalized fluorescence intensities of more than 10 cells. Bars denote standard deviation.

The online version of this article includes the following source data for figure 6:

**Source data 1.** Source data for *Figure 6A,B,C,F,G,H*.

and the diffusion within the nucleolus does not necessarily dominate the exchange with nucleoplasm as would be expected of a prototypical phase-separated condensate.

Treatment with the RNA Pol I inhibitor CX-5461 caused NPM1-GFP to concentrate inside the nucleolar remnants (*Figure 6D* and *Video 6*). In contrast, CDK inhibitor treatment triggered the dispersal of the NPM1-GFP into multiple small globules and increased the diffuse pool in the nucleoplasm (*Figure 6E* and *Video 7*). Although NPM1-GFP globules had the appearance of classic phase-separated droplets, they did not merge over time despite being in close proximity and being quite mobile. The overall amount of NPM1-GFP was not reduced. FRAP analysis of whole nucleoli in RNA Pol I inhibitor-treated cells demonstrated an ~2.5-fold increase in the recovery rate compared to the untreated cells (*Figure 6F and G*), indicating a higher exchange rate of NPM1 molecules between the nucleolar remnants and the nucleoplasm and/or lower nucleolar viscosity. In CDK inhibitor-treated cells, photobleached NPM1-GFP globules recovered even faster ($T_{1/2}$ ~6.9 ± 2.4 s) (*Figure 6F and G*) at a rate comparable to nucleoplasm ($T_{1/2}$ ~6 ± 1.7 s, *Figure 6H*). This rapid exchange rate suggests very weak interactions of NPM1 molecules with the components of these globules. Overall, NPM1 dynamics in RNA Pol I inhibitor-treated cells were consistent with a compromised but extant nucleolar GC layer, while in CDK inhibitor-treated cells they were in line with GC disassembly.

Our interpretation of these results is that nucleolar organization, normal or during stress, is more complex than predicted by multicomponent LLPS alone. Understanding nucleolar stress phenotypes observed in this study from a biophysical perspective will require moving beyond current models that rely solely on phase separation as the basis for nucleolar assembly. Transcriptional activity is strongly correlated with nucleolar integrity and impacts a large number of protein–protein and protein–nucleic acid interactions, some of which occur through specific binding while others are driven by phase separation (*Tartakoff et al., 2022*). The combination of both modes of interaction is likely required for the formation and function of the nucleolus.

## Discussion

Screening a diverse library of chemotherapy drugs has allowed us to identify several compounds that cause changes in nucleolar architecture. This categorization of morphologically distinct nucleolar stresses has provided insights into the biological processes underlying these stresses. Our results show that nucleolar stress can manifest in different forms depending on the biological pathway or pathways targeted by a particular drug (*Figure 7*). The canonical nucleolar stress caused by DNA intercalating agents and manifested by the segregation of stress caps was only one nucleolar stress phenotype. Inhibition of mTOR and PI3K growth pathways resulted in a decrease in nucleolar normality score and rRNA synthesis without dramatic reorganization of nucleolar anatomy. This response may not represent stress per se, but a consequence of the overall downregulation of ribosome biogenesis processes (*Iadevaia et al., 2012*; *Davis et al., 2015*). Inhibitors of HSP90 and the proteasome caused proteotoxic stress – acute loss of protein homeostasis. Accumulation of misfolded and/or undegraded proteins may impair nucleolar functions directly, by clogging its compartments and creating 'aggresome'-based nucleolar stress (*Frottin et al., 2019*), and/or indirectly, by suppressing growth

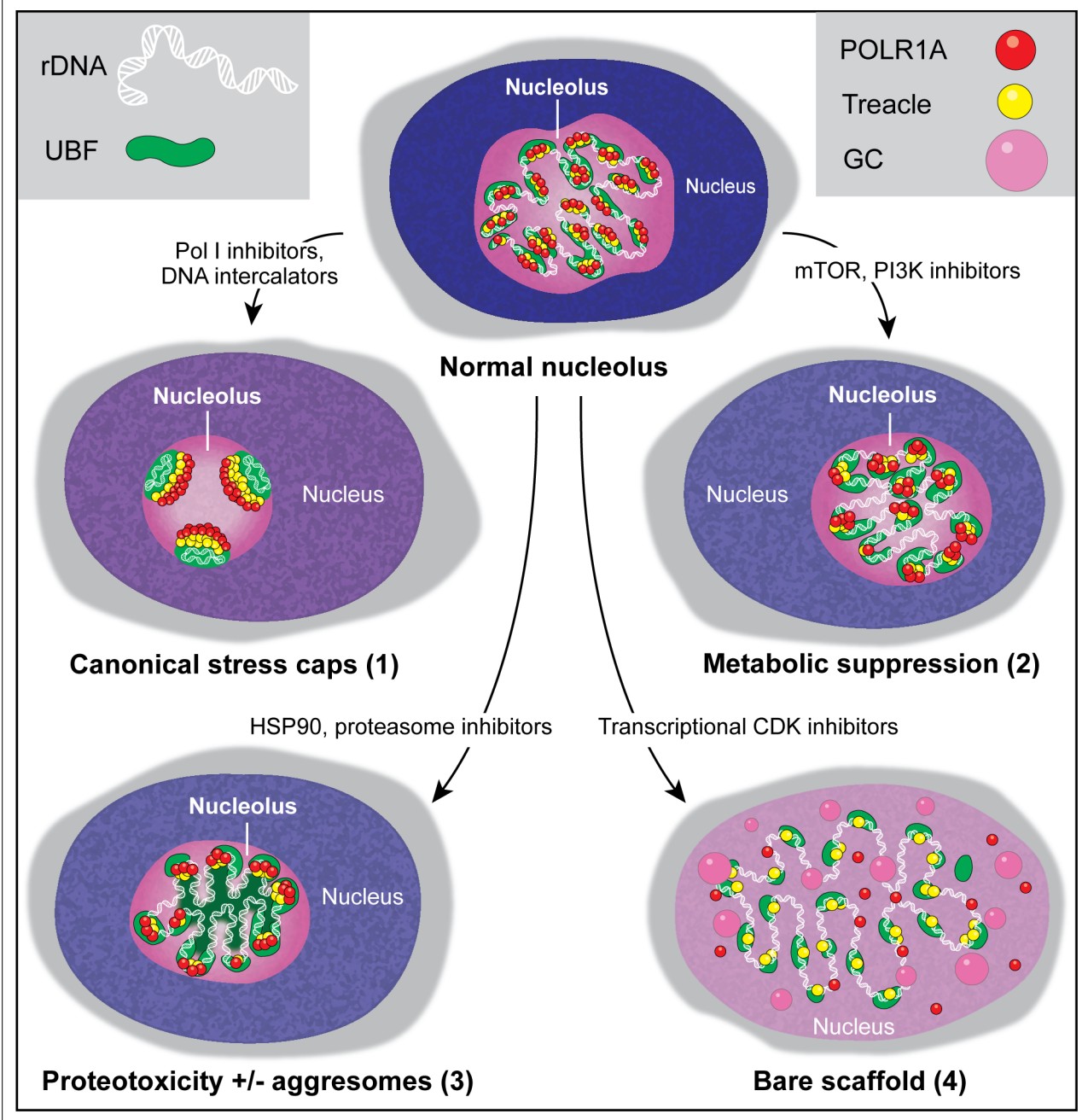

**Figure 7.** Types of nucleolar stresses identified in this study. (1) DNA intercalators and RNA Pol I inhibitors induced canonical nucleolar stress manifested by partial dispersion of granular component (GC) and segregation of rDNA and fibrillar center (FC) components UBF, Treacle, and POLR1A within nucleolar stress caps. (2) Inhibition of mTOR and PI3K growth pathways induced a metabolic suppression of function accompanied by the decrease in nucleolar normality score, size, and rRNA production, without dramatic re-organization of nucleolar anatomy. (3) Inhibitors targeting HSP90 and the proteasome induced proteotoxicity, resulting in the disruption of protein homeostasis and the accumulation of misfolded and/or undegraded proteins. These effects were accompanied by a decrease in nucleolar normality score, rRNA output, and in some cases formation of protein aggregates (aggresomes) inside the nucleolus. (4) Inhibition of transcriptional CDK activity led to the disruption of interactions between rDNA, RNA Pol I, and GC proteins. This resulted in almost complete nucleolar dissolution, leaving behind an extended bare rDNA scaffold with only a few associated FC proteins remaining. UBF and Pol I recruiting protein Treacle remained associated with the rDNA, while POLR1A and GC dispersed in the nucleoplasm. rRNA production ceased and the nucleolar normality score was greatly reduced.

signaling through metabolic pathways (*Su and Dai, 2017*; *Guang et al., 2019*). Nucleoli are often used by pathologists to predict cancer aggressiveness; our studies extend the ability to use nucleolar morphology as a biomarker of underlying cellular state.

The most extreme nucleolar stress in our screen was caused by CDK inhibitors. Rapid nucleolar disintegration implied that constitutive CDK activity is necessary for the assembly of functional RNA Pol I transcriptional complexes and the integrity of the nucleolar compartment. CDK inhibitor flavopiridol was previously shown to impede rRNA production and processing (*Burger et al., 2010*; *Burger et al., 2013*). A recent study attributed the disruptive effect of flavopiridol on RNA Pol I transcription to its inhibition of RNA Pol II (*Abraham et al., 2020*). The absence of nucleolar disruption in cells treated with the catalytic RNA Pol II-specific inhibitor α-amanitin suggests otherwise. Our phosphoproteomics analysis identified multiple nucleolar CDK substrates, arguing that transcriptional CDK activity may affect nucleolar function directly. In vitro phosphorylation of Treacle by CDK9 confirms the kinase specificity at least for this target, which is needed for tethering RNA Pol I machinery on the rDNA. Our findings are consistent with a recent large-scale proteomic study that identified multiple nucleolar proteins as targets of CDK9 (*Johnson et al., 2023*). The idea that transcriptional CDKs drive the RNA production for both RNA Pol I and RNA Pol II offers a coordination mechanism for ribosome biogenesis that requires products of both mRNA and rRNA genes. The same theme of overarching regulation is typical for cell cycle CDKs that drive concerted processes of DNA replication and mitosis by phosphorylating multiple substrates.

There are many existing CDK inhibitors with varying degrees of specificity. Developing inhibitors that target a particular CDK with high specificity is challenging due to the presence of multiple transcriptional CDKs in the human genome that share nearly identical ATP-binding pockets, the sites targeted by ATP-competitive inhibitors (*Jorda et al., 2018*). All CDK inhibitor hits in our screen that had low normality scores were pan-CDK inhibitors. The anticancer compound library also contained CDK inhibitors that were more specific to key cell cycle CDKs: RO3306 (CDK1), BMS265246 (CDK1/2), PD0332991 – palbociclib, LY2835219 – abemaciclib, and LEE011 – ribociclib (CDK4/6). These drugs were not hits in our screen, indicating that the nucleolar stress caused by pan-CDK inhibitors may not be a result of inhibiting cell cycle CDKs. Newer drugs such as AZD4576, JSH-150, and MC180295 are claimed to be specific to the transcriptional CDK9, but it is conceivable that they may have some impact on other transcriptional CDKs. Cyclin-dependent kinases exhibit redundancy in terms of substrate specificity and may compensate for related kinases at least partially. This aspect of CDK biology is better explored for cell cycle CDKs – for instance, mouse and yeast knockout studies showed that the entire cell cycle can run on just CDK1 in the absence of other CDKs (*Kozar and Sicinski, 2005*; *Santamaría et al., 2007*). The ability of related transcriptional CDKs to functionally compensate for each other and/or perform redundant functions has not been well-explored. We allow the possibility that we may be targeting other kinases that can phosphorylate the same nucleolar substrates as CDK9 and their inhibition can cause similar nucleolar stress phenotype. Thus, while our results suggest that CDK9 activity plays a key role in rDNA transcription and nucleolar integrity, it is still possible that other transcriptional CDKs can perform similar functions.

The results of our screen demonstrate that the nucleolus is targeted by many anticancer compounds, whether intentional or not. By combining multiple metrics and approaches, such as live cell imaging, proteomics, biochemistry, rRNA measurements, and immunofluorescence, we highlight the correlation between nucleolar integrity and RNA Pol I transcription in various stressors and provide simple categories that can be used to classify nucleolar stress moving forward. Our biophysical studies emphasize that understanding organizing principles of nucleoli responsible for the diversity of nucleolar stress phenotypes may require integration of both phase separation and affinity models. For drugs that cause nucleolar stress, their antiproliferative activity can be at least in part attributed to disrupting nucleolar processes. Unintended nucleolar stress can also be a source of toxicity. Therapeutic and mechanistic studies should consider nucleolar stress as a potential confounding factor in determining the mechanism of action and toxicity in drug development.

## Materials and methods

### Cell culture, plasmid transfections, and generation of stable cell lines

All cell lines in this study were obtained from ATCC (Manassas, VA) and grown at 37°C in 5% $CO_2$. hTERT RPE1 (cat# CRL-4000), CHON-002 (cat# CRL-2847), and BJ5TA (cat# CRL-4001) were grown in DMEM-F12 medium supplemented with 10% fetal bovine serum (FBS). HCT116 cells were grown in McCoy's 5a modified medium with 10% FBS, DLD1 cells (cat# CCL-221) were grown in RPMI-1640 medium with 10% FBS. All cell lines were mycoplasma negative and authenticated by ATCC STR profiling. Plasmids encoding the human UBF gene tagged with EGFP and nucleolin gene tagged with EGFP were obtained from Addgene (plasmids # 26672 and #28176, respectively) (*Chen and Huang, 2001*; *Takagi et al., 2005*). For generating stable cell lines, RPE1 cells were transfected using X-treme-GENE 9 DNA Transfection Reagent (Roche) according to the manufacturer's directions. Transfected cells were selected with 1 mg/ml G418 (A.G. Scientific).

### Immunofluorescence, high-throughput nucleolar measurements, and calculation of nucleolar normality score

For immunofluorescence, cells were grown on #1.5 glass coverslips, fixed in 4% paraformalde-hyde/PBS for 15 min, and permeabilized with 0.1% Triton X-100. Blocking was done with 5% BSA in PBS/0.1% Triton X-100. Primary and secondary antibodies were diluted in 2.5% BSA/PBS/0.1% Triton X-100. Specimens were incubated with primary antibodies overnight, washed three times for 5–10 min, and incubated with fluorescently conjugated secondary antibodies for 2–4 hr. All washes were performed with PBS/0.5% Triton X-100. DNA was counterstained with DAPI or Hoechst 33342 (Thermo Fisher Scientific). Z-stack images were acquired on the Nikon TiE microscope equipped with a Yokogawa CSU W1 spinning disk and Hamamatsu Flash 4.0 camera using ×60 NA 1.4 or ×100 NA 1.45 objectives.

Calculation of the nucleolar normality score was performed on multichannel single-plane or projection images containing nucleolin, UBF, and DAPI channels, utilizing a custom plugin called '*segment nucleoli jru v4*' (https://github.com/jayunruh/Jay_Plugins3/, copy archived at *Unruh, 2022*). It was written for the open-source image processing program ImageJ (NIH, Bethesda, MD) and is freely available in the Fiji package (*Schindelin et al., 2012*) under the 'Stowers' update site. First, nuclei were segmented based on DAPI labeling. To obtain nuclear masks, the background in a DAPI channel was subtracted with a rolling ball of a large radius, and the resulting image was thresholded at an intensity of ~10% of the image's maximum value. Objects on the edges and outside the size range were excluded. For nucleolar segmentation, the background in the nucleolar channels was subtracted with a small radius rolling ball, and images were smoothened by applying Gaussian blur with a standard deviation of 0.7 pixels. To generate nucleolar masks, UBF signals in each nucleus were thresholded at 40% of the difference between the minimum and maximum values. Objects smaller than 4 pixels were eliminated as noise. The output table contained intensity and area measurements for each nucleolus. All nucleolar measurements were associated with their corresponding nuclei. Nucleoplasmic intensity values were calculated by subtracting the integrated intensity of all nucleoli from the integrated intensity of the whole nuclei. Normality scores for each nucleus were then calculated by dividing the nucleolar/nucleoplasmic ratio of nucleolin by that of UBF:

$$Nucleolar\ normality\ score = (nucleolin \frac{integrated\ nucleolar\ intensity}{integrated\ nucleoplasmic\ intensity})/(UBF \frac{integrated\ nucleolar\ intensity}{integrated\ nucleoplasmic\ intensity})$$

For nucleolar measurements of the nucleolar enrichment of POLR1A, an analogous UBF segmentation strategy was utilized to segment nucleoli within individual nuclei, and POLR1A intensity was determined within and outside nucleolar masks. At least three large fields containing multiple cells were analyzed per condition.

### Anticancer compound library screen, high-throughput imaging, and analysis

The anticancer screening library containing 1180 compounds was purchased from Selleck (cat# L3000-Z304781). For the screen, RPE1 cells were seeded in 384-well plates at 2000 cells/well and incubated

for 4 hr at 37°C. The compound library was added to the cells using a PerkinElmer Janus G3 with a 384w nanohead at a final concentration of 10 µM or 1 µM and incubated for 24 hr. Vehicle-only no-treatment conditions were maintained as controls. Cells were fixed in 4% paraformaldehyde for 15 min and washed/permeabilized in PBS containing 0.1% Triton X100. Liquid handling steps were performed on an Integra Viafill bulk liquid dispenser or by multichannel pipetting. Following fixation and washes using a Biotek 406 washer/dispenser with a 192-pin head, cells were incubated in a blocking solution containing 5% normal goat serum for 1 hr and then with primary antibodies overnight at 4°C. No-primary antibody samples were used as a control for nonspecific secondary antibody binding. Secondary antibodies containing DAPI stain (1 µg/ml final concentration) were applied using the Viafill bulk liquid dispenser and incubated overnight at 4°C. Plates were washed and maintained with PBS at 4°C until imaging. Plates were imaged on an Opera Phenix high-content microscope (PerkinElmer) operated by Harmony High-Content Imaging and Analysis Software 4.9. Images were acquired using a ×40 water objective (NA 1.1). Excitation/emission wavelengths used were 405/435–480 for DAPI, 488/500–550 for Alexa Fluor 488, and 640/650–760 for Cy5. Forty single-plane fields containing hundreds of cells were imaged per well. Images were exported as individual 16-bit TIFF files for processing. High-throughput image processing and calculation of nucleolar normality scores were performed in Fiji. The normality score measurements were performed using '*segment nucleoli jru v4*' plugin as described above. All measurements were aggregated and averages of all fields were calculated for each well. Vehicle controls were highly consistent from plate to plate and were therefore aggregated across all plates to calculate cutoffs for hit selection. Cutoffs for hit calling were set at 2 standard deviations above the average normality score of all DMSO wells.

## 5-EU and Y10b labeling and quantification

For 5-EU incorporation assays, cells were typically seeded in 24-well black optically clear bottom tissue culture-treated plates (Ibidi) and treated with 0.5 mM 5-EU (Thermo Fisher Scientific) for 3 hr. Cells were fixed in ice-cold methanol for 10 min and washed with PBS/0.1% Triton X-100. Fixed cells were stained with 1 µM Alexa Fluor 555 – conjugated azide diluted in PBS containing 2 mM $CuSO_4$ and 50 mM ascorbic acid. To counterstain the DNA, Hoechst 33342 (Sigma) was added to 2 µg/ml. Cells were incubated for several hours or overnight at room temperature protected from light and evaporation, then washed 3× with PBS. For Y10b (anti-rRNA) antibody labeling, cells were grown and fixed as above but processed as regular immunofluorescence. Z-stack images were acquired on the wide-field Nikon Ti2 microscope equipped with Prime95B CMOS camera using ×20 NA 0.5 objective. Image processing was done in Fiji: first, sum intensity Z-projections were generated, then nuclei were segmented on DAPI, and 5-EU or Y10b intensity was measured within nucleolar masks. At least three large fields of view containing hundreds of cells were analyzed to determine the average of each field. The averages of all treatments were normalized to the average of DMSO controls. The final output represents normalized averages of these fields with standard deviation.

## Immuno-FISH

For immuno-FISH assays, cells were grown on #1.5 glass coverslips, fixed in 4% paraformaldehyde in PBS for 15 min, and permeabilized with 0.1% Triton X-100 in PBS. Specimens were then treated with 1 mg/ml RNase A (1:100 from QIAGEN) in PBS for 30 min at 37°C and stored in 25% glycerol/PBS at 4°C. Before hybridization, coverslips were subjected to two freeze-thaw cycles by dipping into liquid nitrogen, treated with 0.1 N HCl for 5 min, washed twice in 2× SSC buffer, and pre-incubated in 50% formamide/2× SSC. Fluorescein-labeled probe for human rDNA (BAC clone RP11-450E20) was obtained from Empire Genomics (Buffalo, NY). Specimens and the probe were denatured together for 7 min at 85°C and hybridized under HybriSlip hybridization cover (GRACE Biolabs) sealed with Cytobond (SciGene) in a humidified chamber at 37°C for 48–72 hr. After hybridization, slides were washed in 50% formamide/2× SSC three times for 5 min per wash at 45°C, then in 1× SSC solution at 45°C for 5 min twice and at room temperature once. Slides were washed again in 0.1% Triton X-100 in PBS and blocked with 5% bovine serum albumin (BSA) in PBS/0.5% Triton X-100. Primary and secondary antibodies were diluted in 2.5% (weight/volume) BSA/PBS/0.1% Triton X-100. Specimens were incubated with primary antibody overnight, washed three times for 5 min, incubated with secondary antibody for several hours, and washed again three times for 5 min. All washes were performed with PBS/0.1% Triton X-100. Vectashield containing DAPI (Vector Laboratories) was used

for mounting. Z-stack images were acquired on the Nikon TiE microscope equipped with a Yokogawa CSU W1 spinning disk and Hamamatsu Flash 4.0 camera using ×100 NA 1.45 objectives.

## NPM1 gene editing and validation

Donor plasmid encoding homology arms and linker-mEGFP sequence for C-terminus tagging of human NPM1 was designed by the Allen Institute for Cell Science and obtained from Addgene (AICSDP-50). The plasmid encoding NPM1-mEGFP was a gift from the Allen Institute for Cell Science (Addgene plasmid # 109122). The sgRNA was synthesized by Synthego with modifications using the proto-spacer sequence UCCAGGCUAUUCAAGAUCUC (*Wienert et al., 2020*). Recombinant high-fidelity *Streptococcus pyogenes* Cas9 HiFi V3 protein was from IDT (cat# 1081061). Cas9 RNP complexes were pre-assembled by mixing 160 pmol Cas9 protein and 140 pmol sgRNA in water and incubated together for 10 min at room temperature. After incubation, 4 µg of donor plasmid DNA was added to the assembly and incubated for an additional 5 min. For electroporation, RPE1 cell pellets containing $1 \times 10^6$ cells were resuspended in Nucleofection Solution for Primary Mammalian Epithelial Cells (Lonza cat# VPI-1005) in the presence of nucleofection enhancer (cat# 1075915) to the total reaction volume of 100 µl. Electroporation was carried out using Amaxa 2b Lonza Nucleofector, program W001. Subsequently, cells were cultured for 7 d, and GFP-positive cells were FACS-sorted into 96-well plates at one cell per well using FACSMelody-Cytometer (BD) operated by FACSDiva 9.1.2 software. Single-cell subclones were expanded, and gene editing was confirmed by fluorescence microscopy, PCR assays for gene insertion with primers outside the homology arms (primer sequences below), and amplicon sequencing by Illumina MiSeq at 250 bp × 250 bp paired-end reads. The resulting sequence data were demultiplexed, followed by an analysis of on-target indel frequency and any expected sequence changes using CRIS.py (1). Selected clones were further validated by western blotting using antibodies against NPM1 and GFP. In addition, cytogenetic analysis was performed on several candidate clones to ensure euploid chromosome number. The edited single-cell subclone used in this study had a correct heterozygous insertion of the mEGFP on the C-terminus of NPM1 and maintained the euploid karyotype (46 chromosomes).

## Primer sequences

Primers around the guide site

> 301-NPM1-ds-F1 cactctttccctacacgacgctcttccgatctaactctctggtggtagaatgaaa
> 301-NPM1-ds-R1 gtgactggagttcagacgtgtgctcttccgatctAACCAAGCAAAGGGTGGA

5′ NPM1-mEGFP junction:

> 301-NPM1-5p-ds-F1 cactctttccctacacgacgctcttccgatctACTTTGGGAGGCAACATGG
> 301-NPM1-5p-ds-R1 gtgactggagttcagacgtgtgctcttccgatctAGGTGTTGGATCACCTGAGA

3′ NPM1-mEGFP junction:

> 301-NPM1-3p-ds-F1 cactctttccctacacgacgctcttccgatctaccagcccggctaatttt
> 301-NPM1-3p-ds-R1 gtgactggagttcagacgtgtgctcttccgatctgagaacattccctcacctactc

Out homology primers:

> NPM1-GFP-OutHA-F2 GCGTGGTAGTTCATGCCTATAA
> NPM1-GFP-OutHA-R2 ACATTCCCTCACCTACTCAAAC

## Live cell imaging, FRAP, and analysis

For live cell imaging, cells were grown on 35 mm ibiTreat µ-dishes (Ibidi, Fitchburg, WI). Time-lapse Z-stack images were captured on a Nikon TiE microscope equipped with ×60 phase-contrast objective NA 1.4, Perfect Focus (PFS) mechanism, Yokogawa CSU-W1 spinning disk, and Flash 4.0 sCMOS camera. Cells were imaged in the regular growth medium; 37°C temperature and 5% $CO_2$ were maintained using an environmental control chamber (Okolab). Images were acquired with the NIS Elements software. Image processing (maximum intensity projection, background subtraction, image registration, and average intensity measurements) was done in Fiji (NIH).

For FRAP, ×100 objective NA 1.45 was used and single-plane images were acquired. GFP was photobleached within a region of interest (ROI) with a pulse of high 488 nm laser power after the initiation of acquisition, and the acquisition continued to monitor the recovery of fluorescence. For FRAP analysis, the background was subtracted using the average intensity value of ROI outside the cell nucleus. The image stack was registered to correct for the cell movement. For every photobleached region, the recovery curve of average intensity was collected, then normalized to the pre-bleach intensity. The average intensity of the whole nucleus was used to correct for photobleaching during the time-lapse acquisition. To calculate $T_{1/2,}$ recovery curves were fit with a two-component exponential recovery function. At least 10 cells were analyzed per condition.

## Phosphoproteomics

For nuclear extracts preparation, RPE1 cells were collected after being treated with 5 μM flavopiridol or 5 μM AZD4573 for 5 hr. Cells were washed with PBS, incubated in hypotonic buffer (0.075 M KCl) containing Halt protease and phosphatase inhibitor cocktail (Thermo) for 10 min at 4°C, and lysed by douncing 10 times. After douncing lysates were spun down for 5 min at 1500 × $g$ at 4°C to collect nuclei. Nuclei were resuspended in a low salt buffer (20 mM HEPES, pH 7.9, 1.5 mM $MgCl_2$, 20 mM KCl, 0.2 mM EDTA, 0.5 mM DTT, protease and phosphatase inhibitor cocktail), followed by addition of an equal volume of a high salt buffer (20 mM HEPES, pH 7.9, 1.5 mM $MgCl_2$, 1.4 M KCl, 0.2 mM EDTA, 0.5 mM DTT, protease and phosphatase inhibitor cocktail). Nuclear proteins were extracted for 30 min at 4°C followed by a 15 min spin at 18,000 × $g$, 4°C. Protein precipitation was carried out by the addition of a trichloroacetic acid (TCA) to a final concentration of 20% and incubation at 4°C overnight. The protein pellet was washed twice with ice-cold acetone and air-dried.

TCA precipitated samples (500 μg) were resuspended in a buffer containing 100 mM Tris-HCl pH 8.5 and 8 M urea. Disulfide bonds were reduced by adding Bond-Breaker TCEP Solution (5 mM final concentration) and incubating at room temperature for 30 min. To prevent bond reformation, chloroacetamide was added (10 mM final concentration) and samples were incubated in the dark for 30 min at room temperature. Proteins were digested with endoproteinase Lys-C (0.4 μg) at 37°C for 6 hr. Samples were diluted with 100 mM Tris-HCl pH 8.5 to reduce the urea concentration to 2 M, $CaCl_2$ was added (2 mM final concentration), and 2 μg trypsin was added to continue the digestion. Samples were then incubated overnight at 37°C. After digestion, the pH of the samples was reduced by adding formic acid (5% final concentration).

Samples were desalted using peptide desalting spin columns (Pierce 89852). After desalting, 10% of each sample was retained for direct mass spectrometry analysis. Phosphopeptides were enriched from the remaining 90% of each sample using the High-Select $TiO_2$ Phosphopeptide Enrichment Kit (Pierce A32993) according to the manufacturer's instructions. Enriched phosphopeptides were resuspended in 25 μl of 0.1% formic acid for mass spectrometry analysis. Proteins were analyzed using Multidimensional Protein Identification Technology (MudPIT). In brief, samples were loaded offline onto three-phase chromatography columns and peptides were eluted using 10 MudPIT steps into an Orbitrap Elite mass spectrometer in positive ion mode (Thermo Scientific) using an Infinity 1260 quaternary pump (Agilent).

RAW files were converted to .ms2 files using RAWDistiller v.1.0. Data were searched using the ProLuCID algorithm version 1.3.5 to match MS/MS spectra to a database containing 44093 human protein sequences (National Center of Biotechnology Information, December 2019 release) and 426 common contaminants, as well as shuffled versions of all sequences (for estimating false discovery rates [FDRs]). Searches were for peptides with static carboxamidomethylation modifications on cysteine residues (+57.02146 Da), for peptides with dynamic oxidation modifications on methionine residues (+15.9949 Da), and for peptides with dynamic phosphorylation modifications on serine, threonine, and tyrosine residues (+79.9663 Da). The in-house software algorithms, swallow and sandmartin, were used in combination with DTASelect and Contrast to filter out inaccurate matches, set protein FDRs below 0.05, and assemble results tables. Proteins were quantified by spectral counting using dNSAF values calculated using NSAF7. Proteins differentially expressed in drug-treated versus untreated cells were determined using the statistical tool QPROT. Proteomics data are available at https://massive.ucsd.edu/ProteoSAFe/static/massive.jsp under the identifier MSV000092420.

## CDK9 kinase assays

Human recombinant Cdk9/cyclin K and CDK9/cyclin T (Sigma-Aldrich) were incubated with the substrate at 30°C for 30 min in reaction buffer (25 mM Tris-acetate [pH 7.9], 10% glycerol, 100 mM KCl, 3 mM DTT) in the presence of 60 µM ATP and 10 µCi gamma-$^{32}$P-ATP. 1 µl (0.1 µg or 0.01 µg) of Cdk9/cyclin K was added per 20 µl reaction. The following substrates were used in reactions: 0.15 µg of recombinant human Treacle (OriGene), and ~25 nM Pol I or Pol II isolated from *S. cerevisiae* (*Appling and Schneider, 2015*). Reactions were halted with an equal volume of SDS protein loading dye. Samples were heated to 95°C for 5 min and loaded into a 5–20% SDS-PAGE gel. After electrophoresis, the gel was wrapped in cellophane and analyzed by phosphoimager (Typhoon 5; GE).

## Treacle immunoprecipitation and western blotting

Cells were collected by spinning down trypsinized cultures at ~200 × *g* for 5 min at 4°C; trypsin was neutralized by the addition of FBS before centrifugation. Cell pellets were washed with ice-cold PBS and lysed in ice-cold IP lysis buffer (Pierce 87787) supplemented with Halt Protease and Phosphatase Inhibitor Cocktail (Thermo Fisher Scientific) for 30 min and dounced 10 times. Lysates were cleared by centrifugation at 16,000 × *g* for 10 min at 4°C. Rabbit anti-Treacle antibody was bound to protein A Dynabeads (Thermo Fisher Scientific) for 30 min and washed in PBS with 0.1% Tween-20. Cell lysates containing an equivalent amount of total protein were incubated with Dynabeads conjugated to Treacle-antibody for 3 hr at 4°C, shaking. Dynabeads were washed three times using IP lysis buffer and re-suspended in RIPA buffer (Thermo Fisher Scientific). Proteins were eluted by the addition of NuPAGE LDS Sample Buffer (Thermo Fisher Scientific) containing 5% beta-mercaptoethanol (Sigma) and boiling for 10 min. Protein samples were separated by SDS-PAGE in 4–12% Bis-Tris gels (Thermo Fisher Scientific), transferred to PVDF membrane, blocked in SuperBlock (TBS) Blocking Buffer (Thermo Fisher Scientific), and washed with TBST. Primary antibodies were detected using horseradish peroxidase-conjugated secondary antibodies and developed using the WesternBright (Advansta) detection kit. Chemiluminescence was detected using G:Box Chemi XT4 (Syngene).

## Antibodies used in this study

Anti-UBF #H00007343-M01, Abnova
Anti-UBF NBP1-82545, Novus Biologicals
Anti-Nucleolin #ab70493; Abcam
Anti-Ki-67 #9449; Cell Signaling Technology
Anti-rRNA (Y10b) #sc-33678, Santa Cruz Biotechnology
Anti-Treacle (TCOF1) #ab224544, Abcam
Anti-POLR1A (RPA 194, C-1) #sc-48385, Santa Cruz Biotechnology
Anti-POLR1A (RPA 194, F-6) #sc-46699, Santa Cruz Biotechnology
Anti-β-actin #3700; Cell Signaling Technology

Secondary antibodies for immunofluorescence (Alexa 488, 555, and 647 conjugates) were obtained from Life Technologies and used at 1:500 dilution. Secondary HRP-conjugated antibodies for western blotting were from Cell Signaling Technology and typically used at 1:5000 dilution.

## Acknowledgements

We thank Tissue Culture and Microscopy core facilities at the Stowers Institute for enabling many of our experiments. We are grateful to Kym Delventhal, Brandon Miller, and Kyle Weaver from Genomic Engineering core facility and Kevin Ferro from Flow Cytometry core facility for their help with generating Cas-9 edited NPM1-eGFP RPE1 cell line. We are thankful to Lauren Weems and Ella Leslie from the Screening core facility for their help with the drug screen. We thank members of Gerton lab for the discussions. We are grateful to Mark Miller for help with illustrations. This study was supported by R35-GM140710 to David A Schneider and by funding from the Stowers Institute for Medical Research.

# Additional information

### Funding

| Funder | Grant reference number | Author |
|---|---|---|
| National Institute of General Medical Sciences | R35 GM140710 | David Alan Schneider |

The funders had no role in study design, data collection and interpretation, or the decision to submit the work for publication.

### Author contributions

Tamara A Potapova, Conceptualization, Formal analysis, Validation, Investigation, Visualization, Methodology, Writing - original draft, Writing - review and editing; Jay R Unruh, Data curation, Software, Formal analysis, Validation, Writing - original draft; Juliana Conkright-Fincham, Investigation, Methodology; Charles AS Banks, Conceptualization, Investigation, Methodology; Laurence Florens, Conceptualization; David Alan Schneider, Conceptualization, Funding acquisition, Investigation, Methodology; Jennifer L Gerton, Conceptualization, Resources, Funding acquisition, Investigation, Writing - original draft, Project administration, Writing - review and editing

### Author ORCIDs

Tamara A Potapova ⓘ http://orcid.org/0000-0003-2761-1795
Jay R Unruh ⓘ http://orcid.org/0000-0003-3077-4990
Laurence Florens ⓘ http://orcid.org/0000-0002-9310-6650
David Alan Schneider ⓘ http://orcid.org/0000-0003-0635-5091
Jennifer L Gerton ⓘ http://orcid.org/0000-0003-0743-3637

Joint Public Review: https://doi.org/10.7554/eLife.88799.3.sa1
Author Response https://doi.org/10.7554/eLife.88799.3.sa2

---

# Additional files

### Supplementary files

• Supplementary file 1. The complete list of anticancer compounds present in the library with normality scores after 1 µM and 10 µM treatments. Normality scores for DMSO controls are also listed.

• Supplementary file 2. List of nuclear proteins selectively enriched and depleted in cells treated with CDK inhibitors. QPROT statistics for all identified proteins are also provided.

• Supplementary file 3. List of differentially phosphorylated nuclear proteins in cells treated with CDK inhibitors. The complete list of phosphopeptides with sites is also provided.

• MDAR checklist

### Data availability

Original data underlying this manuscript can be accessed from the Stowers Original Data Repository at http://www.stowers.org/research/publications/libpb-2415. Custom FIJI plugins are available at https://github.com/jayunruh/Jay_Plugins3/, (copy archived at *Unruh, 2022*). Proteomics data are available at https://massive.ucsd.edu/ProteoSAFe/static/massive.jsp under the identifier MSV000092420.

The following datasets were generated:

| Author(s) | Year | Dataset title | Dataset URL | Database and Identifier |
|---|---|---|---|---|
| Florens L | 2023 | Distinct states of nucleolar stress induced by anti-cancer drugs | https://massive.ucsd.edu/ProteoSAFe/dataset.jsp?accession=MSV000092420 | MassIVE, MSV000092420 |
| Potapova TA, Unruh JR, Conkright-Fincham J, Banks CAS, Florens L, Schneider DA, Gerton JL | 2023 | Distinct states of nucleolar stress induced by anti-cancer drugs | https://www.stowers.org/research/publications/libpb-2415 | Stowers Original Data Repository, libpb-2415 |

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
