## [Editor Report · eLife assessment]

This study and associated data is **compelling**, novel, **important**, and well-carried out. The study demonstrates a novel finding that different chemotherapeutic agents can induce nucleolar stress, which manifests with varying cellular and molecular characteristics. The study also proposes a mechanism for how a novel type of nucleolar stress driven by CDK inhibitors may be regulated. The study sheds light on the importance of nucleolar stress in defining the on-target and off-target effects of chemotherapy in normal and cancer cells.

---

## [Referee Report · Joint Public Review]

Summary:

This is an interesting study with high quality imaging and quantitative data. The authors devise a robust quantitative parameter that is easy applicable to any experimental system. The drug screen data can potentially be helpful to the wider community studying nucleolar architecture and effects of chemotherapy drugs. Additionally, the authors find Treacle phosphorylation as a potential link between CDK9 inhibition, rDNA transcription and nucleolar stress. Therefore I think this would be of broad interest to researchers studying transcription, CDKs, nucleolus and chemotherapy drug mechanisms.

Revised manuscript:

While most of my concerns related were addressed, a PolI ChIP on rDNA would be an important experiment to establish the relevance of some of the conclusions of the paper using well established protocols with validated antibodies for PolI ChIP. Furthermore, additional S to A mutants of Treacle S1299A/S1301A is an important control which could have provided evidence if indeed S1299/S1301 were the only sites being phosphorylated by CDK9. To support their model, the authors should test if overexpression of Treacle mutants S1299A/S1301A can partially phenocopy the nucleolar stress seen upon CDK9 inhibition. This would considerably strengthen the author's claim that reduced Treacle phosphorylation leads to Pol I disassociation from rDNA and consequently leads to nucleolar stress. If not, it would have strengthened the authors' argument that Treacle could have multiple sites targeted by CDK9 and that mutating any one or two may not be sufficient to cause disassociation from PolI.

Overall, I believe the primary conclusions regarding the impact of various chemotherapy drugs on nucleolar state are solid and valuable to the broader scientific community. However, the mechanistic exploration of CDK9i is not sufficiently developed, and the authors have not adequately addressed the feedback provided in the original manuscript.

---

## [Author Response]

The following is the authors’ response to the original reviews.

**eLife assessment**
This study and associated data is compelling, novel, important, and well-carried out. The study demonstrates a novel finding that different chemotherapeutic agents can induce nucleolar stress, which manifests with varying cellular and molecular characteristics. The study also proposes a mechanism for how a novel type of nucleolar stress driven by CDK inhibitors may be regulated. The study sheds light on the importance of nucleolar stress in defining the on-target and offtarget effects of chemotherapy in normal and cancer cells.

We are thankful to the reviewers and the editor for their feedback and thorough assessment of our work. Our responses to the comments and suggestions are below.

**Reviewer #1 (Public Review):**
The study titled "Distinct states of nucleolar stress induced by anti-cancer drugs" by Potapova and colleagues demonstrates that different chemotherapeutic agents can induce nucleolar stress, which manifests with varying cellular and molecular characteristics. The study also proposes a mechanism for how a novel type of nucleolar stress driven by CDK inhibitors may be regulated. As a reviewer, I appreciate the unbiased screening approach and I am enthusiastic about the novel insights into cell biology and the implications for cancer research and treatment. The study has several significant strengths: (i) it highlights the understudied role of nucleolar stress in the on- and off-target effects of chemotherapy; (ii) it defines novel molecular and cellular characteristics of the different types of nucleolar stress phenotypes; (iii) it proposes novel modes of action for well-known drugs. However, there are several important points that should be addressed:• The rationale behind choosing RPE cells for the screen is unclear. It might be more informative to use cancer cells to study the effects of chemotherapeutic agents. Alternatively, were RPE cells selected to evaluate the side effects of these agents on normal cells? Clarifying these points in the introduction and discussion would guide the reader.

RPE1, a non-cancer-derived cell line, was chosen for this study to evaluate the effects of anticancer drugs on normal nucleolar function, with the underlying premise that nucleolar stress in normal cells can contribute to non-specific toxicity. This clarification is added to the introduction. Another factor that played in selecting a normal cell line for the drug screen and subsequent experiments was the spectrum of known and unknown genetic and metabolic alterations present in various cancer cell lines. These variables are often unique to a particular cancer cell line and may or may not impact nucleolar proteome and function. Therefore, the nucleolar stress response can be influenced by the spectrum of alterations inherent to each cancer. Our primary focus was to determine the impact of these drugs under normal conditions.

That said, the selected hits of main drug classes were validated in a panel of cell lines that included two other hTERT lines (BJ5TA and CHON-002) and two cancer lines (DLD1 and HCT116). In cancer cells starting nucleolar normality scores were lower than in hTERT cells, suggesting that genetic and metabolic changes in these cells may indeed affect nucleolar morphology. Nonetheless, all drugs from a panel of selected hits from different target classes validated in both cancer cell lines (Fig. 2F).

• Figure 2F indicates that DLD1 and HCT116 cells are less sensitive to nucleolar changes induced by several inhibitors, including CDK inhibitors. It would be crucial to correlate these differences with cell viability. Are these differences due to cell-type sensitivity or variations in intracellular drug levels? Assessing cell viability and intracellular drug concentration for the same drugs and cells would provide valuable insights.

One of the reasons for the reduced magnitude of the effects of selected drugs in DLD1 and HCT116 cells is their lower baseline normality scores compared to hTERT cells (now shown in Sup. Fig. 1B-C). Other potential factors include proteomic and metabolic shifts and alterations in signaling pathways that control ribosome production. The less-likely possibility of variations in intracellular drug levels cannot be excluded, but measuring this for every compound in every cell line was not feasible in this study. These limitations are now noted in the results section.

Regarding the point about viability - our initial screen output, in addition to normality scores, included cell count (cumulative count of cells in all imaged fields), which serves as a proxy for viability. By this measure, all hit compounds in our screen were cytostatic or cytotoxic in RPE1 cells (Fig. 2C). The impact of these drugs on the viability of cancer cells that can have various degrees of addiction to ribosome biogenesis merits a separate study of a large cancer cell line panel.

• Have the authors interpreted nucleolar stress as the primary cause of cell death induced by these drugs? When cells treated with CDK inhibitors exhibit the dissociated nucleoli phenotype, is this effect reversible? Is this phenotype indicative of cell death commitment? Conducting a washout experiment to measure the recovery of nucleolar function and cell viability would address these questions.

Whether nucleolar toxicity is the primary cause of cytotoxicity for a given chemotherapy drug is an incisive and thought-provoking question. Our screen did not discern whether the cytotoxic effects of our hits were due to inhibition of their intended targets, their impact on the nucleolus, or a combined effect. This point is now mentioned in the results section. Regarding the reversibility of the nucleolar disassembly phenotype seen in CDK inhibitors –in the case of flavopiridol, which is a reversible CDK inhibitor, we demonstrated that nucleoli re-assembled within 4-6 hours after the drug was washed out. An example of this is shown in Sup. Figure 3 and in Video 5. For these experiments, cells were pretreated with the drug for 5 hours, not long enough to cause cell death.

• The correlation between the loss of Treacle phosphorylation and nucleolar stress upon CDK inhibition is intriguing. However, it remains unclear how these two events are related. Would Treacle knockdown yield the same nucleolar phenotype as CDK inhibition? Moreover, would point mutations that abolish Treacle phosphorylation prevent its interaction with Pol-I? Experiments addressing these questions would enhance our understanding of the correlation/causation between Treacle phosphorylation and the effects of CDK inhibition on nucleolar stress.

We agree that the Treacle finding is interesting and warrants further investigation. In our attempts to knock down Treacle with siRNA, its protein levels were reduced by no more than 50%, which was not sufficient to cause a strong nucleolar stress response. Therefore, these data were not incorporated into the manuscript. However, in our view, Treacle is unlikely to be the only nucleolar CDK substrate whose dephosphorylation is causing the “bare scaffold” phenotype caused by the transcriptional CDK inhibitors. Our phospho-proteomics studies identified multiple nucleolar CDK substrates with established roles in the formation of the nucleolus. For instance, the granular component protein Ki-67 was also dephosphorylated on multiple sites and dispersed throughout the nucleus (shown in Sup. Fig 4). Given that CDKs typically phosphorylate many substrates that can have multiple phosphorylation sites, identifying a sole protein or phosphorylation site responsible for nucleolar disassembly may be an unattainable target.

Overall, this study is significant and novel as it sheds light on the importance of nucleolar stress in defining the on-target and off-target effects of chemotherapy in normal and cancer cells.

Thank you, we appreciate the positive and constructive assessment of our study.

**Reviewer #2 (Public Review):**
This is an interesting study with high-quality imaging and quantitative data. The authors devise a robust quantitative parameter that is easily applicable to any experimental system. The drug screen data can potentially be helpful to the wider community studying nucleolar architecture and the effects of chemotherapy drugs. Additionally, the authors find Treacle phosphorylation as a potential link between CDK9 inhibition, rDNA transcription, and nucleolar stress. Therefore I think this would be of broad interest to researchers studying transcription, CDKs, nucleolus, and chemotherapy drug mechanisms. However, the study has several weaknesses in its current form as outlined below.1. Overall the study seems to suffer from a lack of focus. At first, it feels like a descriptive study aimed at characterizing the effect of chemotherapy drugs on the nucleolar state. But then the authors dive into the mechanism of CDK inhibition and then suddenly switch to studying biophysical properties of nucleolus using NPM1. Figure 6 does not enhance the story in any way; on the contrary, the findings from Fig. 6 are inconclusive and therefore could lead to some confusion.

This study was specifically designed to examine a broad range of chemotherapy drugs. The newly created nucleolar normality score enabled us to measure nucleolar stress precisely and in high throughput. Our primary objective was to find drugs that disrupt the normal nucleolar morphology and then study in-depth the most interesting and novel hits. We have made revisions to emphasize that these are the primary focal points of the manuscript.

As context, we were motivated to explore the biophysical properties of the nucleolus because they are thought to underlie its formation and function, which also suggested a potential predictive value for modeling nucleolar responses to drug treatments. For this, we edited the RPE1 cell line by endogenously tagging NPM1, a granular component protein that behaves in line with the phase-separation paradigm in vitro and when over-expressed. We fully expected to confirm that its behavior in vivo would be consistent with LLPS, but instead found that even in an untreated scenario, the dynamics of endogenous NPM1 could not be fully explained by the phase separation theory (Fig. 6 A-C). Our message is that accurately predicting drug responses using the nucleolar normality score as a readout, based on our current understanding of the biophysical forces governing nucleolar assembly, is unworkable. For instance, normality scores decrease and NPM1 dynamics increase radically when CDKs are inhibited, without changes in NPM1 concentration or concentrations of other protein components (Fig.6 E-H). These observations are important because they highlight our gaps in understanding the relative contribution of phase separation versus active assembly in nucleolar formation. We believe that these observations are worth sharing with the scientific community.

1. The justification for pursuing CDK inhibitors is not clear. Some of the top hits in the screen were mTOR, PI3K, HSP90, Topoisomerases, but the authors fail to properly justify why they chose CDKi over other inhibitors.

We decided to focus on CDK inhibitors for several reasons. First, their effects were completely new and unexpected, suggesting the existence of an unknown mechanism regulating nucleolar structure and function. In addition, CDK inhibitors caused a very strong and distinct nucleolar stress phenotype with the lowest normality scores that merited its own term, the “bare scaffold” phenotype. One more reason for pursuing CDK-inhibiting drugs was their high rate of failure in clinics because of the intense and hard-to-explain toxicity. We suspect that this toxicity may be due at least in part to their profound effect on nucleolar organization and ribosome production throughout the body. We stated this rationale more explicitly in the manuscript.

1. In addition to poor justification, it seems like a very superficial attempt at deciphering the mechanism of CDK9imediated nucleolar stress. I think the most interesting part of the study is the link between CDK9, Pol I transcription, and nucleolar stress. But the data presented is not entirely convincing. There are several important controls missing as detailed below.

We agree with the reviewer that follow-up studies of CDK9, Pol I, and nucleolar stress connection are important long-term goals. However, the primary objective of this study was to ascertain the scope of anticancer agents that can cause nucleolar stress and the establishment of nucleolar stress categories. This is an important advance and could serve as the foundation for a standalone in-depth study or multiple studies. We have included the complete screen, proteomics, and phospho-proteomics results (Sup. Tables 1, 2, and 3), which will enable other investigators to mine the screen information based on their specific interests. Furthermore, we have made multiple text revisions to clarify rationale and interpretation, and incorporated additional data that strengthen the manuscript.

1. The authors did not test if inhibition of CDK7 and/or CDK12 also induces nucleolar stress. CDK7 and CDK12 are also major kinases of RNAPII CTD, just like CDK9. Importantly, there are well-established inhibitors against both these kinases. It is not clear from the text whether these inhibitors were included in the screen library.

Our anticancer compound library contained CDK7 inhibitor THZ1⦁2HCL, and it was a hit at both 1 and 10 uM concentrations (Sup. Table 1). However, its nucleolar stress phenotype was morphologically distinct from CDK9 inhibitors, resembling the stress caps phenotype instead of the bare scaffold phenotype. We did not pursue CDK7 because of its two hard-to-separate functions: in addition to its role as an RNAPII CTD kinase, it also acts as a CDK-activating kinase (CAK) by promoting the associations of multiple CDKs with their cyclin partners. This dual role of CDK7 makes the interpretation of THZ1-induced nucleolar stress phenotype difficult because it could be attributed to either or both of these functions. Moreover, it was reported to cause DNA damage, which may explain why it causes stress caps. An image depicting nucleolar stress phenotype caused by THZ1⦁2HCL is provided in Author response image 1.

**Author response image 1. sa2fig1:** Control and THZ1 - treated RPE1 cells, images from screen plates.

We are not aware of specific inhibitors of CDK12, as they also reportedly inhibit CDK13. None of the CDK12/CDK13 inhibitors were present in our library, therefore we can neither confirm nor exclude the possible involvement of these kinases in regulating nucleolar structure. Many other existing CDK inhibitors were absent from our library. Our work highlights the importance of assessing their potential to induce nucleolar stress and offers an approach for this assessment.

1. In Figure 4E, the authors show that Pol I is reduced in nucleolus/on rDNA. The authors should include an orthogonal method like chromatin fractionation and/or ChIP

We acknowledge the reviewer’s request for additional validation of reduced occupancy of rDNA by Pol I.

Nucleolar chromatin fractionation in cells treated with CDK inhibitors is unlikely to work due to nearly complete nucleolar disassembly. Chromatin immunoprecipitation would require finding and validating a suitable ChIP-grade antibody. Moreover, the evaluation of repetitive regions by ChIP is non-trivial and error-prone. To help address this request and further confirm the POLR1A immunofluorescence results in 4E, we included additional immunofluorescence data obtained with a different POLR1A antibody (Sup. Fig. 3D), and the results were similar.

1. In Fig. 5D, in vitro kinase lacks important controls. The authors should include S to A mutants of Treacle S1299A/S1301A to demonstrate that CDK9 phosphorylates these two residues specifically.1. To support their model, the authors should test if overexpression of Treacle mutants S1299A/S1301A can partially phenocopy the nucleolar stress seen upon CDK9 inhibition. This would considerably strengthen the author's claim that reduced Treacle phosphorylation leads to Pol I disassociation from rDNA and consequently leads to nucleolar stress.1. Additionally, it would be interesting if S1299D/S1301D mutants could partially rescue CDK9 inhibition.

Points (6-8):

We reiterate that transcriptional CDKs target multiple nucleolar proteins, and the observed phenotype might be due to the combined effects of de-phosphorylation of multiple substrates. We concur that deconstructing the role of Treacle phosphorylation sites is very interesting and warrants further in-depth studies. The phospho-proteomics enrichment method, while an effective first-pass strategy, might not capture 100% of the phosphorylated sites. Treacle is a phospho-protein with an abundance of serine and threonine residues. It could potentially have been selectively dephosphorylated on more sites than were detected by this method. Therefore, the suggested mutations may not be the exclusive contributors responsible for the functional phenotype. Additionally, overexpressing Treacle impairs the viability of RPE1 cells, complicating the interpretation of experiments involving overexpression of both wild-type and mutant proteins. A conceivable strategy would involve generating phosphomimetic and non-phosphorylatable mutants by gene editing, studying their interactions by biochemical approaches, and determining their impact on nucleolar function, but this may take years of additional work. We hope that our work will inspire further studies that explore Treacle phosphorylation and other functions of transcriptional CDKs in nucleolar formation.

Thank you for the thoughtful review and suggestions.

**Reviewer #2 (Recommendations For The Authors):**
1. The manuscript could be re-organized to focus on 'CDK9-Treacle-Pol I-nucleolar stress' as the central part of the story.

While we acknowledge this suggestion, it's important to emphasize that the primary focus of this manuscript is on the identification of anticancer drugs that induce nucleolar stress and the establishment of nucleolar stress categories.

1. Include a "no ATP" control in the in vitro kinase assay and indicate molecular sizes.

We provided an additional kinase assay (Sup. Fig. 4B) that includes no ATP control lanes and a fragment of a Coomassie blue stained gel showing molecular weight markers. No ATP control assays (lanes 4 and 5) were blank as expected. Molecular weight markers were added to all other kinase assays based on the known sizes of isolated Pol II holoenzyme subunits Rbp1 (191 kDa) and Rbp2 (138 kDa).

1. For in vitro phosphorylation, please provide an explanation for using CDK9/cyclin K instead of Cyclin T1 which is the predominant cyclin for CDK9

Recombinant CDK9/cyclin K complex was used for in vitro kinase assays for a technical reason: CDK9/cyclin T obtained from the same vendor appeared to be low quality, as it showed only minimal activity toward our positive control, the isolated Pol II complex. The kinase assays using recombinant CDK9/cyclin T in parallel with CDK9/cyclin K are now presented it Sup. Fig. 4B. The first two assays in this experiment contained Pol II as a substrate, and it is evident that Pol II was phosphorylated much stronger by CDK9/cyclin K than CDK9/cyclin T (comparing lane 1 vs lane 2). Therefore, the lack of detectable Treacle phosphorylation by CDK9/Cyclin T (lane 7), in contrast to strong phosphorylation by CDK9/cyclin K (lane 6), was likely attributable to poor reagent quality rather than physiological differences. We can conclude that CDK9/cyclin K reliably phosphorylates Treacle in vitro, but CDK9/cyclin T kinase assays were inconclusive.